# The Automated LLM Speedrunning Benchmark: Reproducing NanoGPT Improvements

**Bingchen Zhao**[★1,2]   **Despoina Magka**[★1]   **Minqi Jiang**[★1]

**Xian Li**[1]   **Roberta Raileanu**[1]   **Tatiana Shavrina**[1]   **Jean-Christophe Gagnon-Audet**[1]

**Kelvin Niu**[1]   **Shagun Sodhani**[1]   **Michael Shvartsman**[1]   **Andrei Lupu**[1]

**Alisia Lupidi**[1]   **Edan Toledo**[1]   **Karen Hambardzumyan**[1]   **Martin Josifoski**[1]

**Thomas Foster**[1]   **Lucia Cipolina-Kun**[1]   **Abhishek Charnalia**[1]   **Derek Dunfield**[1]

**Alexander H. Miller**[1]   **Oisin Mac Aodha**[2]   **Jakob Foerster**[1]   **Yoram Bachrach**[1]

[★]Equal contribution

[1]Meta Superintelligence Labs   [2]University of Edinburgh

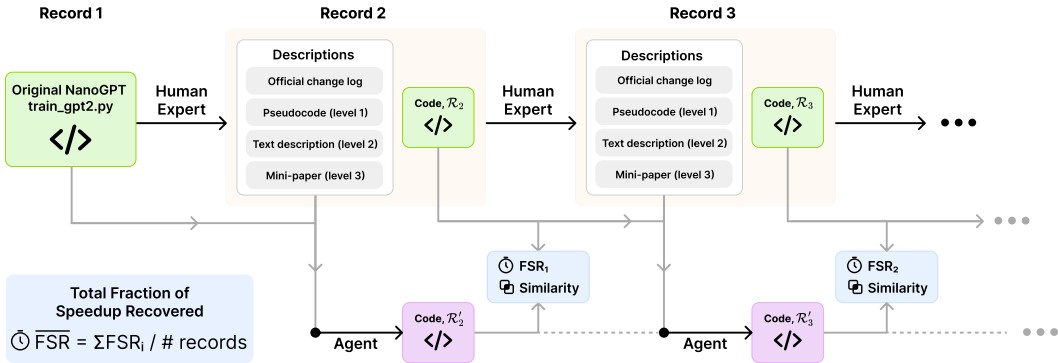

Figure 1: The Automated LLM Speedrunning Benchmark. We create a task for each consecutive pair of records $\mathcal{R}_i$, $\mathcal{R}_{i+1}$. The performance of the agent is evaluated by comparing the relative speedup of the agent solution $\mathcal{R}'_i$ to $\mathcal{R}_i$.

## Abstract

Rapidly improving large language models (LLMs) have the potential to assist in scientific progress. One critical skill in this endeavor is the ability to faithfully reproduce existing work. To evaluate the capability of AI agents to reproduce complex code in an active research area, we introduce the Automated LLM Speedrunning Benchmark, leveraging the research community's contributions to the *NanoGPT speedrun*, a competition to train a GPT-2 model in the shortest time. Each of the 19 speedrun tasks provides the agent with the previous record's training script, optionally paired with one of three hint formats, ranging from pseudocode to paper-like descriptions of the new record's improvements. Records execute quickly by design and speedrun improvements encompass diverse code-level changes, ranging from high-level algorithmic advancements to hardware-aware optimizations. These features make the benchmark both accessible and realistic for the frontier problem of improving LLM training. We find that recent frontier reasoning LLMs combined with SoTA scaffolds struggle to reimplement already-known innovations in our benchmark, even when given detailed hints. Our benchmark thus provides a simple, non-saturated measure of an LLM's ability to automate scientific reproduction, a necessary (but not sufficient) skill for an autonomous research agent. Rapid advancements in large language models (LLMs) have the potential to assist in

scientific progress. A critical capability toward this endeavor is the ability to reproduce existing work. To evaluate the ability of AI agents to reproduce results in an active research area, we introduce the Automated LLM Speedrunning Benchmark, leveraging the research community's contributions on the *NanoGPT speedrun*, a competition to train a GPT-2 model in the shortest time. Each of the 19 speedrun tasks provides the agent with the previous record's training script, optionally paired with one of three hint formats, ranging from pseudocode to paper-like descriptions of the new record's improvements. Records execute quickly by design and speedrun improvements encompass diverse code-level changes, ranging from high-level algorithmic advancements to hardware-aware optimizations. These features make the benchmark both accessible and realistic for the frontier problem of improving LLM training. We find that recent reasoning LLMs combined with SoTA scaffolds struggle to reimplement already-known innovations in our benchmark, even when given detailed hints. Our benchmark thus provides a simple, non-saturated measure of an LLM's ability to automate scientific reproduction, a necessary (but not sufficient) skill for an autonomous research agent.

# 1    Introduction

The advent of LLMs capable of succeeding in challenging math, coding, and scientific reasoning domains has led to a surge of activity in applying LLM agents to the longstanding ambition of automated scientific discovery [Simon, 1995, Langley, 1987, Waltz and Buchanan, 2009, King et al., 2009, Steinruecken et al., 2019]. Early results suggest LLM-based systems can improve the productivity of human researchers, from formulating hypotheses to implementing code-based experiments to testing them [Romera-Paredes et al., 2024, Castro et al., 2025, Yin, 2025, Inizan et al., 2025].

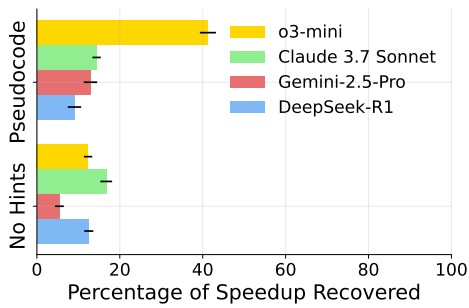

Figure 2: Recent LLM agents struggle to reproduce NanoGPT Speedrun records.

Scientific progress hinges on trustworthy results, and the ultimate test of the truth behind a finding is whether the experiment and its outcomes can be reproduced [Fineberg et al., 2019, Pineau et al., 2021, Henderson et al., 2018]. Thus, a critical component of automated science is *automated reproducibility*: the process of automatically reimplementing an experiment based on a description of the experiment design, such that the implementation reproduces previously reported outcomes. In other words, translating the description of an experiment into its implementation [Peng, 2011, Siegel et al., 2024]. Moreover, success in reimplementing a known study also serves as a metric for assessing the reliability with which an agent can implement experiments via description, an ability that would enable researchers to quickly scale up the testing of new ideas, regardless of whether they are of human or AI origin.

We study the ability of recent reasoning LLMs in combination with state-of-the-art *scaffolds*—programs that iteratively make use of an LLM for finding a solution to a given task—on reproducing prior discoveries in the domain of LLM training. We henceforth refer to the combination of a specific LLM and scaffold for the purpose of automated research as a *research agent*, and use the more specific term *AI research agent* to refer to those specifically designed for automating AI research itself. While there is much speculation that AI research agents may lead to the beginnings of a recursive self-improvement loop for future LLM-based research agents, we set our focus on the more modest goal of understanding whether current AI research agents can succeed at the prerequisite task of reproducing previous scientific findings on GPT-2 [Radford et al., 2019], the first model to demonstrate a broad capacity for zero-shot transfer to new tasks via prompting.

Towards this goal, we introduce *The Automated LLM Speedrunning Benchmark*, based on the series of community-driven improvements to GPT-2 training in the *NanoGPT Speedrun* [Jordan et al., 2024a], a competition based on minimizing the wall time of training an open-source PyTorch reimplementation of GPT-2 [Karpathy, 2023] to reach a target cross-entropy loss of 3.28 on the validation set of FineWeb [Penedo et al., 2024], using a single 8×H100 node. Since its inception in June 2024, this community effort has driven the training time of GPT-2 from 45 minutes to below 3 minutes (as of

Table 1: Key motivations of our benchmark design and how it differentiates from existing ML reproducibility benchmarks. Here, "Reproducibility" denotes whether the tasks require replicating a given technique; "Sequential", whether the benchmark measures reproducibility over a cumulative series of scientific results; "LLM research", whether the task involves language model development; and "Agent scaffold", whether a baseline agent scaffold is released with the benchmark.

| | Reproducibility | Sequential | LLM research | Agent scaffold |
|---|---|---|---|---|
| MLE-bench [Chan et al., 2025] | No | No | No | No |
| PaperBench [Starace et al., 2025] | Yes | No | Partially | Yes |
| CORE-bench [Siegel et al., 2024] | Yes | No | No | Yes |
| RE-bench [Wijk et al., 2024] | No | No | Yes | Yes |
| MLAgentBench [Huang et al., 2024] | No | No | Partially | Yes |
| MLGym-bench [Nathani et al., 2025] | No | No | Partially | Yes |
| **Automated LLM Speedrunning** (ours) | Yes | Yes | Yes | Yes |

May 2025). These improvements were driven by new algorithmic enhancements, some of which have been shown to generalize beyond the scale of the 124M parameter GPT-2 model, with the most notable being the invention of the Muon optimizer [Jordan et al., 2024b], later demonstrated to show benefits for training much larger modern LLMs [Liu et al., 2025a, Shah et al., 2025]. Other speedrun improvements include mixed precision training and more efficient attention variants [Dong et al., 2024]. As of May 2025, the NanoGPT Speedrun includes 21 successive speedrun records. Each record is associated with its corresponding training script (`train_gpt.py`), a measured training time, a public announcement of the changes, and a high-level summary of the code changes.[1]

The Automated LLM Speedrunning Benchmark then tasks an AI research agent with reproducing each successive speedrun record, starting from the previous record, with an optional set of hints of various formats and levels of detail. The clear code-level ground-truth targets per record alongside detailed change logs between records make this benchmark an ideal testing ground for the ability of agents to reproduce not only a single experimental finding, but also a series of cumulative research findings—a distinct affordance compared to prior reproducibility benchmarks. Here, all tasks share the same success metric of training time to reach the target validation loss, measured on a fixed hardware configuration (a single 8xH100 node), making exact reproduction, fair comparisons, and cross-task comparisons straightforward. Lastly, perhaps the most compelling aspect of this benchmark is its focus on reproducing discoveries directly relevant to real-world LLM development.

Our experiments show that even when given a description of the difference between two consecutive speedrun records in various formats, recent agents based on DeepSeek-R1 [DeepSeek-AI et al., 2025] and o3-mini [OpenAI, 2025] combined with a state-of-the-art search scaffold, still struggle to improve ground-truth records to match the speedups of the next ground-truth record (see Figure 2).

We believe the Automated LLM Speedrunning Benchmark can spur development of AI research agents that can automate reproducibility studies, paving a critical step on the way towards more capable AI research agents that can realize the aspiration of accelerating the pace of scientific discovery via automated science. However, our results show that before such lofty goals can be realized, automated reproducibility remains a central challenge that must be addressed.

## 2   Related works

**Automated reproducibility.** Recent works have devised benchmarks for evaluating the ability of LLM agents to reproduce code-based experiments from published papers. CORE-Bench measures an agent's ability to correctly install, execute, and interpret a paper's associated codebase and its outputs [Siegel et al., 2024]. Other benchmarks, including PaperBench [Starace et al., 2025], Papers2Code [Seo et al., 2025], AutoP2C [Lin et al., 2025], and SciReplicate [Xiang et al., 2025] test the agent's ability to convert a research paper to a codebase that replicates the reported findings or the agent's ability to formulate and test hypotheses [Chen et al., 2025, Liu et al., 2025b]. Instead of evaluating on a wide set of, often, unrelated papers as in these previous works, the Automated LLM Speedrunning Benchmark focuses on a single important overarching task of speeding up LLM training. This focus allows for a unified success metric across a diverse gradation of task complexity,

---

[1] https://github.com/KellerJordan/modded-nanogpt?tab=readme-ov-file#world-record-history

defined by the natural path of innovation previously discovered by human researchers. This grounding allows for not only comparison to granular, ground-truth code-level changes, but also opens the door to evaluating an LLM agent's ability to reproduce an entire research arc over multiple compounding innovations against human performance. Moreover, the benchmark's multiple hint levels allow for controlled study of how performance varies across different forms of background information.

**Code generation with LLMs.** Code is inherently reproducible via repeated execution and requires no additional equipment to run beyond a computer. Thus, many automated scientific reproducibility benchmarks, including ours, focus primarily on virtual, code-based experiments. In this domain, research agents directly benefit from and build upon the rapid progress in coding and computer-use agents, such as a growing set of complex, sandboxed software-engineering agent benchmarks [Yang et al., 2024, Wang et al., 2024, Fourney et al., 2024, Mialon et al., 2023, Yoran et al., 2024, Zhou et al., 2023, Koh et al., 2024] and scaffold designs [Zhang et al., 2024], such as AIDE [Jiang et al., 2025], which we both use as a baseline and extend in our experiments.

**LLMs for automated ML.** Recent advances enabling LLMs to exploit chain-of-thought outputs during inference have led to drastic improvements in their performance on reasoning tasks in domains like math, coding, and science. These improvements have led to a surge in LLM programs seeking to automate the key parts of machine learning itself, encompassing iterated hypothesis generation and testing and the writing of reports detailing the findings, in the form of end-to-end agents [Lu et al., 2024, Huang et al., 2025, Yamada et al., 2025a], agents focused on hypothesis generation [Gottweis et al., 2025, O'Neill et al., 2025], as well as agents that can interact with a human-in-the-loop to jointly formulate and test hypotheses [Intology AI, 2025, Autoscience Institute, 2025]. However, early results suggest these systems, while capable of optimizing code-level improvements, often fall short in executing on experiments that faithfully reflect their intended goals [Yamada et al., 2025b]. Thus, while LLM-based reasoning models can generate, at times, novel hypotheses [Gu et al., 2024], their ability for scientific reproduction remains a crucial bottleneck in automating scientific research.

# 3 The Automated LLM Speedrunning Benchmark

The Automated LLM Speedrunning Benchmark seeks to evaluate an LLM agent's ability to reproduce the wall-time speedup associated with each record transition from the NanoGPT Speedrun, both with and without access to hints describing the corresponding changes at varying levels of abstraction. Table 1 summarizes how our work compares to existing ML reproducibility benchmarks.

## 3.1 Reproducibility tasks from existing records

For each transition from record $\mathcal{R}_{i-1}$ to record $\mathcal{R}_i$ for $i = 2, ..., 21$, excluding $i = 7$, whose speedup is purely due to upgrading PyTorch, we define the following components:

$\mathcal{R}_i$   Training script for the $i$-th record in the speedrun,

$t_i$   Wall-clock time (in seconds) required by $\mathcal{R}_i$ to reach the target validation loss,

$\Delta_i^1$   Level 1 hint: A *pseudocode* description of code change from the previous record,

$\Delta_i^2$   Level 2 hint: A *natural-language* description of the code change from the previous record,

$\Delta_i^3$   Level 3 hint: A *mini-paper* summarizing the code change from the previous record.

All hints were first drafted by R1, manually verified, and, where necessary, edited for correctness and relevance. See Appendix F for further details on our hint creation process. We provide a categorized listing of all ground-truth records in Appendix G and example hints in Appendix H.

For convenience, we denote the set of ground-truth speedrun records (which excludes record 6) as $\mathcal{I}$. We define a *record task* as a tuple $\langle \mathcal{R}_{i-1}, \mathcal{R}_i, t_i, m \rangle$, where $\mathcal{R}_1$ corresponds to the initial NanoGPT training script, and where $m$ is any subset of the set of *hint levels*, $\{0, 1, 2, 3\}$, where level 0 corresponds to *no hint*. Depending on the presence of hints, we categorize the possible tasks in our benchmark into two types:

**Record reproduction tasks.** Given hints that describe the subsequent record, i.e. $m \neq \{0\}$, the LLM agent must reproduce record $\mathcal{R}_{i+1}$ given $\mathcal{R}_i$ and the set of corresponding hints. Here the key metric of interest is the *fraction of speedup recovered* (FSR), defined as

$$\text{FSR}_i = \frac{t_i - t'_{i+1}}{t_i - t_{i+1}}. \tag{1}$$

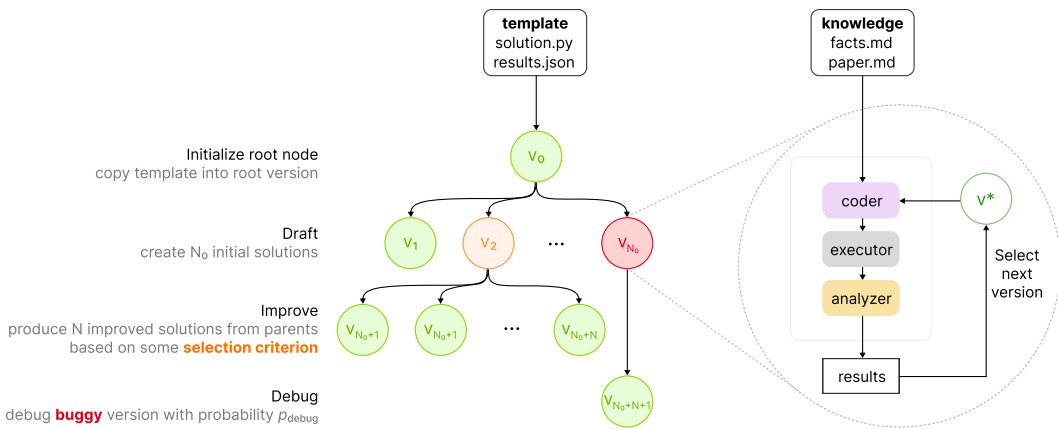

Figure 3: Overview of our flexible search scaffold. Search starts from a root node containing code for the starting record $\mathcal{R}_i$ from which $N_0$ initial solutions are generated. Subsequently, each search iteration debugs a buggy leaf node with probability $p_{\text{debug}}$ and otherwise greedily selects the best node to improve, with debug and improvement each branching $N$ solutions. At each search step, the coder submodule implements the solution, with optional access to external knowledge (e.g. hints).

where $t'_{i+1}$ is the training time achieved by the agent to reach the target validation loss. The full benchmark performance is then the mean FSR over the set of all included records, $\mathcal{I}$:

$$\overline{\text{FSR}} = \frac{1}{|\mathcal{I}|} \sum_i \frac{t_i - t'_{i+1}}{t_i - t_{i+1}}. \tag{2}$$

**Record optimization tasks.** Without any hints, i.e. $m = \{0\}$, the LLM agent must produce a new training script solution $\mathcal{R}'_{i+1}$ with a minimal training time $t'_{i+1}$ to reach the target validation loss, given $\mathcal{R}_i$. Here we consider both the raw wall time $t'_{i+1}$ of the solution produced, in addition to $\text{FSR}_i$. Similar to the setting of record reproduction, we consider the mean of these metrics over all ground-truth records in the benchmark as an overall measure of performance. This allows the agent to explore its own improvements given the same SoTA starting point that humans had when each record was produced.

## 3.2 Agent scaffolds

We provide a flexible search scaffold implementation that extends AIDE [Jiang et al., 2025] into a more general parameterization. In this setup, visualized in Figure 3, each node in the search tree represents a solution instance contained in a directory with relevant scripts, performance metrics, and an LLM-generated execution summary. For instance, in NanoGPT training, a solution node consists of a single `train_gpt2.py` script and a results file describing its performance and execution outcome. The fitness of each node is evaluated based on these metrics—such as wall time to reach the target validation loss—with each new search initialized using a ground-truth script from the benchmark and proceeding by branching into up-to-multiple child solutions.

Each search step follows three stages: implementation, execution, and analysis. During implementation, the agent generates working code from a prompt that includes the task description and optionally, a set of associated hints. We use Aider [Gauthier, 2025] to make diff-based edits to the initial solution, producing modified versions for execution. These solutions are then run on an 8xH100 node, and the output is summarized in natural language via the analysis stage, capturing key performance indicators and insights from standard outputs. Custom prompts guide each stage and are detailed in Appendix F. The search begins with $N_0$ initial modifications to the root node. At each step, a new node branches from either a randomly chosen buggy node (with probability $p_{\text{debug}}$) or the highest-performing node. To avoid redundant debugging, we cap retries at $D_{\text{max}}$ per node. This scaffold design supports multiple search variants, outlined in Table 2, with each receiving the same budget $M$ of search steps to ensure fair comparison.

Table 2: Search variants and their corresponding scaffold parameterizations.

| Method | Initial branch factor | Branch factor | Debug probability | Max debug depth |
|--------|-----------------------|---------------|-------------------|-----------------|
| Tree | 1 | $N$ | 0 | 0 |
| Forest | $N_0$ | $N$ | 0 | 0 |
| AIDE | $N_0$ | 1 | $p_{\text{debug}}$ | $D_{\text{max}}$ |
| Multi-AIDE | $N_0$ | $N$ | $p_{\text{debug}}$ | $D_{\text{max}}$ |
| Flat (Best-of-M) | $M$ | — | — | — |

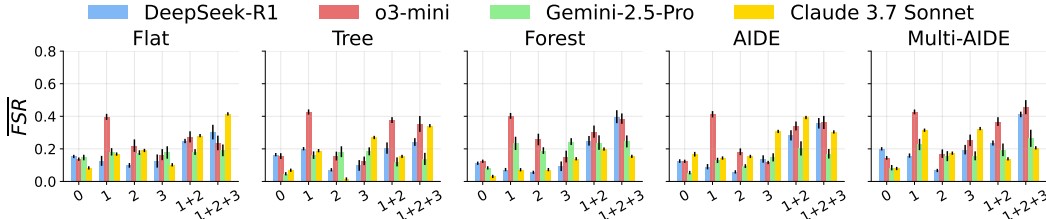

Figure 4: Mean FSR across five search variants and four frontier models for six hint regimes: no hint (0), pseudocode (1), text (2), mini-paper (3) and combinations thereof $(1+2, 1+2+3)$.

# 4 Experiments and results

We now evaluate the performance of several baseline agents across a range of scaffolds, hint formats, and model backbones for all NanoGPT Speedrun records. We report results using the normalized runtime improvement metric (FSR) from Equation 2, as well as measures of code similarity between agent and human solutions. For fair comparisons, we use training times for human records based on rerunning each ground-truth record on the same hardware configuration as agent solutions. Appendix C reports the near exact reproduction of training times for human records on our cluster.

## 4.1 Baselines

We compare a number of LLM agents based on DeepSeek-R1, o3-mini, Gemini-2.5-Pro, and Claude-3.7-Sonnet, using instances of the search scaffolds listed in Table 2. Our choice of parameters are $N_0 = 3$ for the initial pool of root hypotheses (forest, AIDE and multi-AIDE), $N = 3$ for the branching factor (tree, forest and multi-AIDE), $p_{\text{debug}} = 0.5$ and $D_{\text{max}} = 5$ for the debug probability and maximum debug depth respectively (AIDE and multi-AIDE), and a search budget of $M = 20$ nodes. Taken together, these scaffolds cover a range of branching factors, search depth, and debug logic.

For each pair of model and search scaffold, we assess the mean FSR across all 19 tasks for each of the following hint levels: no hint (level 0), pseudocode (level 1), text description (level 2), and mini-paper (level 3). Each solution is executed under a maximum runtime of 60 minutes (i.e. a maximum of 20 hours per agent run). We observe an average run time of $\approx$10 hours per agent run, across a total of 6,840 agent runs (19 records × 6 hint regimes × 5 search variants × 4 models × 3 seeds), for a total of 6,840 × 8 H100 (internal cluster) hours spent executing the generated solutions.

## 4.2 Reproducing individual records

We report the mean FSR for each model, search scaffold, and hint-level combination across 3 full search runs in Figure 4, including the case of no hints. It is evident that hints are necessary for inducing greater values of FSR, with all agents failing to recover more than 20% of the speed-up achieved by human solutions on average without hints. Appendix D further reports the mean FSR for each individual record transitions per agent variation across 3 runs per variation.

We observe that o3-mini generally achieves equal or better results than other models in mean FSR for all hint levels, but sees slightly worse performance with no hints. Notably, flat search (i.e. best-of-M), generally matches or outperforms iterated search scaffolds across the individual hint levels (levels 1–3), while matching their performance in the case of no hints. Moreover, tree and forest methods, which lack debug steps, perform on par with AIDE-based search scaffolds, suggesting that explicit debug steps do not provide a significant benefit on top of iterative improvement steps. Overall, the

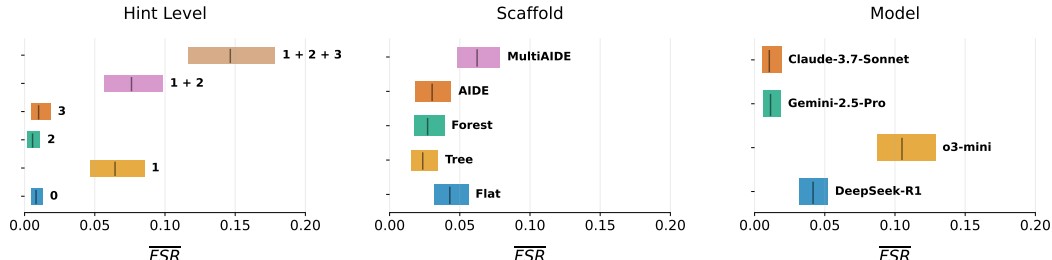

Figure 5: Interquartile Mean (IQM) evaluation results. Scores are aggregated across multiple runs with the same hint level, scaffold, and model.

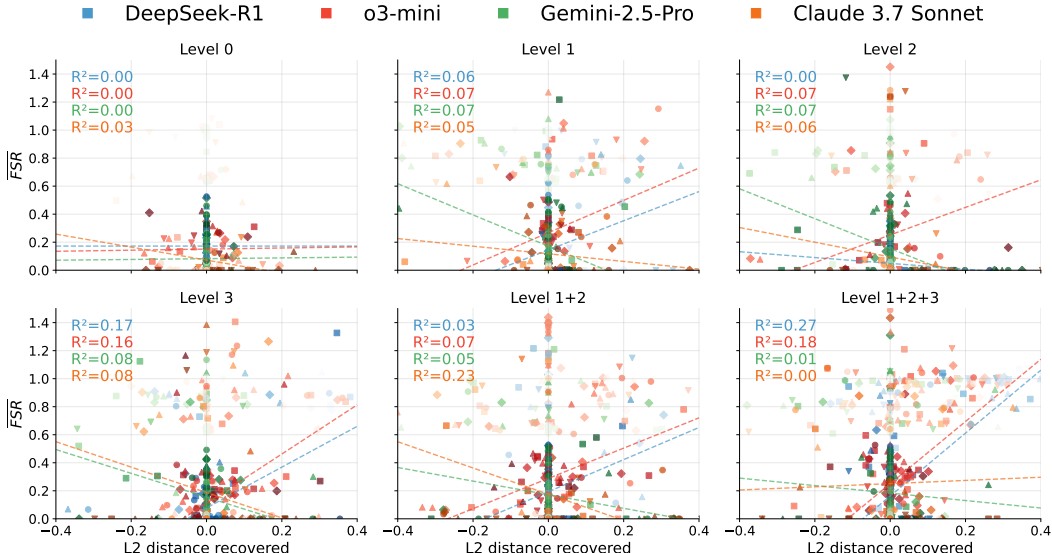

Figure 7: Correlation of FSR with L2 distance recovered for each hint level, showing a modest correlation between similarity to the human solution and FSR for most hint types and models.

gap between the best models (o3-mini and Claude-3.7-Sonnet) and the open-weights (R1) is wider for the search scaffolds incorporating branching logic (tree, search, and AIDE variants), suggesting that models like o3-mini can better iterate on their previous solutions. Figure 6 further shows how agents tend to have more difficulty in reproducing later records.

Out of the various hint formats, the most useful are the pseudocode and the combinations of pseudocode with text and mini-papers hints, which enable o3-mini to recover approximately 40% and 46%, respectively, of the speed-up attained by human solutions on average. Surprisingly, R1 agents seem to worsen with the presence of the individual hints, generally achieving lower FSR compared to the no-hint setting, suggesting that attempting to implement the complex changes in these hints results in buggy code. With hints, R1 produces solutions with lower FSR than simply making no changes to the code, a common outcome with no hints, as indicated by the cluster around a recovered L2 embedding distance of 0.0 in Figure 7 (Section 4.6 details this similarity analysis).

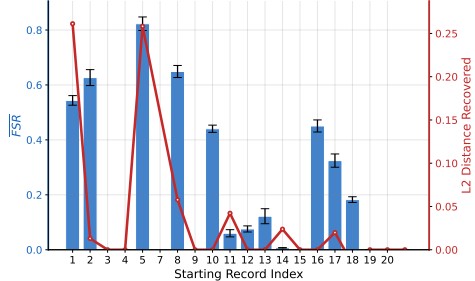

Figure 6: FSR and embedding distance per record for o3-mini with text description hints (mean and std over 3 seeds). Later records tend to be harder for agents, leading to lower recovered embedding distance and speedups.

### 4.3 Combining multiple hints

We further investigate the impact of combining hint formats, and also include these results for each agent variation in Figure 4. We observe that providing the text description or mini-paper together

Table 3: Performance comparison across different hint formats (mean and std over 3 runs). Color-coded values are differences relative to the best-performing individual hint in the combination.

| Hints | Model | Flat | Tree | Forest | AIDE | Multi-AIDE |
|---|---|---|---|---|---|---|
| L1 (pseudocode) | o3-mini | **0.40**±0.02 | **0.43**±0.02 | **0.40**±0.02 | **0.41**±0.02 | 0.43±0.02 |
| L2 (text) | o3-mini | 0.22±0.04 | 0.16±0.03 | 0.26±0.04 | 0.18±0.02 | 0.17±0.03 |
| L3 (mini-paper) | o3-mini | 0.17±0.03 | 0.13±0.03 | 0.15±0.04 | 0.12±0.01 | 0.25±0.04 |
| L1+L2 | o3-mini | 0.27±0.03 (-0.13) | 0.38±0.02 (-0.05) | 0.31±0.04 (-0.09) | 0.34±0.03 (-0.07) | 0.37±0.03 (-0.06) |
| L1+L2+L3 | o3-mini | 0.24±0.05 (-0.16) | 0.35±0.05 (-0.08) | 0.39±0.03 (-0.01) | 0.36±0.04 (-0.05) | **0.46**±0.04 (+0.03) |
| L1 (pseudocode) | DeepSeek-R1 | 0.13±0.03 | 0.20±0.00 | 0.07±0.00 | 0.09±0.02 | 0.16±0.01 |
| L2 (text) | DeepSeek-R1 | 0.10±0.01 | 0.07±0.00 | 0.06±0.00 | 0.06±0.01 | 0.07±0.00 |
| L3 (mini-paper) | DeepSeek-R1 | 0.13±0.04 | 0.10±0.03 | 0.09±0.03 | 0.14±0.02 | 0.20±0.03 |
| L1+L2 | DeepSeek-R1 | 0.25±0.01 (+0.12) | 0.20±0.03 (+0.00) | 0.25±0.03 (+0.18) | 0.28±0.03 (+0.19) | 0.24±0.02 (+0.08) |
| L1+L2+L3 | DeepSeek-R1 | **0.30**±0.04 (+0.17) | **0.24**±0.02 (+0.04) | **0.40**±0.04 (+0.31) | **0.36**±0.03 (+0.22) | **0.41**±0.02 (+0.21) |
| L1 (pseudocode) | Gemini-2.5-Pro | 0.18±0.02 | 0.16±0.02 | 0.23±0.04 | 0.13±0.02 | 0.23±0.03 |
| L2 (text) | Gemini-2.5-Pro | 0.18±0.01 | 0.18±0.03 | 0.19±0.02 | 0.09±0.01 | 0.16±0.03 |
| L3 (mini-paper) | Gemini-2.5-Pro | 0.18±0.04 | **0.18**±0.02 | 0.24±0.02 | 0.15±0.02 | 0.16±0.03 |
| L1+L2 | Gemini-2.5-Pro | 0.18±0.02 (+0.00) | 0.12±0.03 (-0.06) | 0.24±0.04 (+0.01) | **0.20**±0.04 (+0.07) | 0.19±0.04 (-0.04) |
| L1+L2+L3 | Gemini-2.5-Pro | **0.19**±0.04 (+0.01) | 0.14±0.04 (-0.04) | **0.25**±0.04 (+0.01) | 0.17±0.03 (+0.02) | **0.26**±0.05 (+0.03) |
| L1 (pseudocode) | Claude-3.7-Sonnet | 0.14±0.03 | 0.13±0.03 | 0.05±0.01 | 0.14±0.01 | 0.18±0.04 |
| L2 (text) | Claude-3.7-Sonnet | 0.10±0.03 | 0.03±0.01 | 0.06±0.02 | 0.14±0.02 | 0.14±0.02 |
| L3 (mini-paper) | Claude-3.7-Sonnet | 0.06±0.02 | 0.22±0.02 | 0.11±0.01 | **0.34**±0.01 | 0.19±0.03 |
| L1+L2 | Claude-3.7-Sonnet | 0.14±0.03 (+0.00) | 0.11±0.02 (-0.11) | **0.15**±0.02 (+0.04) | 0.30±0.02 (-0.04) | 0.09±0.01 (-0.09) |
| L1+L2+L3 | Claude-3.7-Sonnet | **0.21**±0.04 (+0.07) | **0.31**±0.02 (+0.09) | 0.10±0.02 (-0.01) | 0.31±0.01 (-0.03) | **0.20**±0.02 (+0.01) |

with the pseudocode compared to only providing the pseudocode hint can substantially degrade performance for o3-mini (see o3-mini result in Table 3), but surprisingly benefits R1. These results suggest that o3-mini may be less capable of taking advantage of longer contexts, while R1's reasoning directly benefits from longer initial prompts. On the other hand, the effect of combined hints on Gemini-2.5-Pro and Claude-3.7-Sonnet appears relatively small, suggesting they can handle longer context yet lacks the ability to leverage them for effective reasoning for reproducing code changes.

## 4.4 Interquartile mean evaluation

As the agent runs could bring a large variance in the experimental results, in Figure 5 we present the aggregated Interquartile Mean (IQM) results across runs with the same hint level, search scaffold, and model. The IQM metric has been shown to be robust to comparisons with a small sample size, and in Figure 5 we report as $95\%$ confidence intervals, bootstrapped from 3 seeds following Agarwal et al. [2021]. On the hint level comparison, the agents reach the best performance when using all three hints combined. For individual hints, the pseudo-code hint performs the best. For search scaffold, multi-AIDE search outperforms all others. On the model side, we are surprised to find that the Gemini-2.5-Pro and Claude-3.7-Sonnet gives the lowest performance close to 0 FSR, even lagging behind the open-sourced R1 model. The results also suggest Automated LLM Speedrunning is a challenging benchmark for current agents as the aggregated performances are fairly low.

## 4.5 Analysis of search trees

To better understand how each agent spends its search budget, we inspect the proportion of different kinds of nodes in their search trees: buggy nodes, which crash due to runtime errors; improved nodes, which successfully improved runtime compared to their parent; and unimproved nodes, which do not improve from their parent. This breakdown of the search trees is presented in Figure 8. We observe that flat search leads to a higher total proportion of buggy nodes, indicating that initially-proposed solutions are most often incorrect. We also notice that R1 agents generate more buggy nodes under AIDE and multi-AIDE—the two variants with debugging steps—suggesting that R1 may be less capable of fixing its own mistakes compared to o3-mini. Gemini-2.5-Pro tends to generate fewer buggy nodes compared to the other models, yet it lags behind on the FSR metric (see Figure 4 and Figure 2), suggesting that Gemini produces more robust code at the cost of correctly implementing the more efficient solutions described in the hints. Surprisingly, Claude-3.7-Sonnet generates significantly more buggy nodes than the other three models, with the fraction of buggy nodes gradually overtaking the fraction of working nodes in the search tree, indicating that Claude-3.7-Sonnet struggles to improve and debug its previous solutions.

The analysis of node types in the search tree provides insight into the discrepancy on the results of Claude-3.7-Sonnet between the $\overline{FSR}$ results in Figure 4 and the IQM results in Figure 5. While the

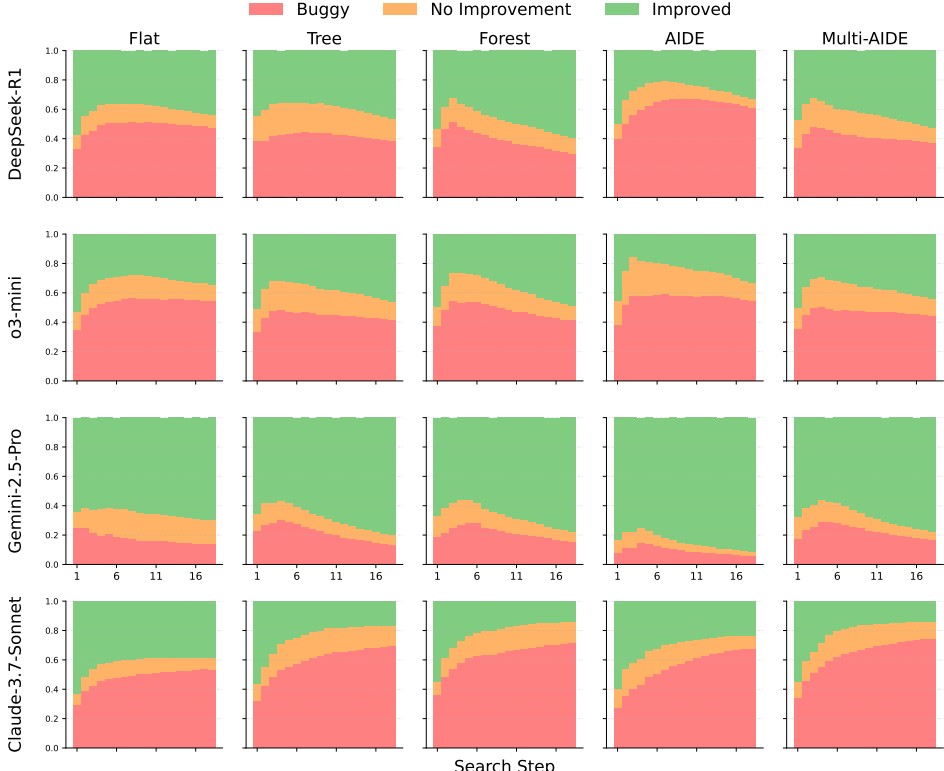

Figure 8: Fraction of node types across search trees for each model and search method. Notably, branching (i.e. non-flat) search is beneficial for reducing the proportion of buggy nodes. Further, a majority of non-buggy steps produce improved nodes for all branching search methods.

$\overline{FSR}$ results suggest that, on average, Claude-3.7-Sonnet performs comparably to o3-mini, the IQM plot indicates that Claude-3.7-Sonnet significantly lags behind o3-mini. This discrepancy can be explained by examining the distribution of node types in the search tree. Claude-3.7-Sonnet is capable of generating working solutions that substantially improve the $FSR$. However, it also produces a considerable number of buggy nodes that result in runtime errors. These errors negatively impact the overall performance as reflected in the IQM plot, despite the improvements in the averaged $\overline{FSR}$.

## 4.6 Similarity between agent and human solutions

Agents may output solutions with similar performance to human ones, but may still fail to reproduce the target code changes. We thus assess code similarity between agent and human solutions by comparing code embedding distances using the SFR-Embedding-Code 2B model [Liu et al., 2024].

Specifically, we normalize the embedding distance between the agent's code solution and the target human solution, i.e. the next record, and divide this distance by the embedding distance between the current and the next human record. Figure 7 depicts the normalized L2 embedding distance recovered with respect to the record speedups and for each type of hint. Here the distance recovered is defined as $1 - \|e_{i+1} - e'_{i+1}\| / \|e_{i+1} - e_i\|$, where $e_i$ is the embedding for $\mathcal{R}_i$, and $e'_i$ is the embedding for the LLM's attempt at reproducing it. We observe a stronger correlation between higher similarity score and FSR for richer hint formats, suggesting that distances under this embedding space can be a meaningful measure of degree of successful reproduction.

As an alternative measure of code similarity, we made use of R1 as a judge, prompting it to assess what fraction of the ground-truth code changes between the current and next record were successfully reproduced in the agent's solution, on a scale of 0 to 1 with a score of 1 corresponding to a completely correct reimplementation. Appendix E contains a comparison between these judge-based similarity scores and FSR across all agent variations. We observe clear positive correlation between higher similarity scores and FSR. We provide sample outputs from R1 judge in Appendix F.

# 5 Limitations and future directions

Our Automated LLM Speedrunning Benchmark serves as a challenging evaluation of an LLM agent's ability to reproduce scientific findings specific to LLM training. However, there remain important limits in its capacity for assessing an agent's true capability for scientific reproduction, and each of these limitations point the way to directions for exciting future research.

**Scaling up external knowledge.** By design, the various hint levels are succinct and easily fit within the context of the LLMs we tested. Moreover, these hints were manually defined, with the relevant hint directly provided as part of the associated task instance. A more realistic setup would provide the agent with the ability to use external knowledge via some form of function calling, including the ability to store intermediate results in various kinds of memory structures, e.g. a short-term scratchpad, long-term database, or neural module [Hermann et al., 2015, Weston et al., 2014]. Accessing a wider and potentially accumulating set of external information would also test the agent's ability to manage information whose total size may exceed its context length [Sarthi et al., 2024].

**Memorization or generalization?** As many of the ground-truth records in the NanoGPT Speedrun were published potentially before the cut-off date of the models used in our experiments (and thus, most likely of future models), there is the possibility that models may have already seen these solutions during training [Gupta and Pruthi, 2025]. We find that neither R1 nor o3-mini accurately reproduce the speedups realized in the ground-truth records, but explicitly disentangling memorization from generalization may become more necessary as models begin to saturate the benchmark. More advanced techniques for measuring memorization in LLMs would allow for a more nuanced evaluation [Carlini et al., 2021, Razeghi et al., 2022, Oren et al., 2023, Deng et al., 2024].

**Semantic diffs.** Our experiment analysis focuses on FSR and numeric similarity scores between the LLM's solution and the corresponding human solution. Moving beyond a similarity score toward more expressive natural-language summaries, e.g. via automatically-generated commit messages [Jiang et al., 2017], of the code diffs between LLM and human solutions would allow for more scalable identification of common mistakes or new innovations with respect to the human solutions.

**From LLM speedrun to ML speedrun.** The skills required for the LLM speedrun are a good starting point but are not enough to create reliable AI research agents. True research agents must handle more complex tasks, such as working with entire multi-file codebases, optimizing for metrics beyond training time like model performance or memory usage, dealing with distributed training, and defining their own success metrics. Most importantly, the current benchmark tests the ability to reproduce results, not to innovate. While an LLM beating human records would be a milestone, the ultimate test is whether future agents can solve new, open scientific challenges.

# 6 Conclusions

We introduced the Automated LLM Speedrunning Benchmark, a challenging evaluation of an LLM agent's ability to reproduce existing scientific innovations in LLM training, based on reproducing each successive record in the community-driven NanoGPT Speedrun. Unlike previous benchmarks for automated scientific reproducibility, our benchmark enables evaluations of an agent's ability to reproduce not just a single result, but each incremental advance across a chain of research innovations. We found that even recent, leading reasoning models, like R1 and o3-mini, when combined with a state-of-the-art agent scaffold, still struggle to successfully produce speedrun solutions that match the speedups attained by the corresponding human solutions. Moreover, this gap between human and agent performance persists even when these strong baseline agents are provided with detailed explanations describing the exact code changes from the previous speedrun record. Our results suggest that automated reproducibility may serve as a significant obstacle in realizing reliable, autonomous research agents with current, leading models, and we expand on the potential societal impacts of our work in Appendix I. We believe the Automated LLM Speedrunning Benchmark can be an effective testbed for monitoring this crucial capability in future research agents.

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

# Appendix

## A  Experimenting with additional knowledge

Certain records are particularly challenging for our baseline agents, such as $\mathcal{R}_{12}$, which achieves its speedup via FlexAttention [Dong et al., 2024], a PyTorch module that enables performant implementation of custom attention variants and was released in August 2024, potentially after the knowledge cut-off of R1 and o3-mini.

Table A.1: FSR of $\mathcal{R}'_{12}$ worsens when FlexAttention docs are inserted in the model's context.

| $\mathcal{R}'_{12}$ | DeepSeek-R1 | o3-mini |
|---|---|---|
| with docs | 0.07±0.01 | 0.06±0.01 |
| without docs | **0.09±0.01** | **0.10±0.01** |

To determine whether the agents' poor performance on $\mathcal{R}_{12}$ was due to missing in-weights knowledge of this module, we inserted content from the blog post describing FlexAttention (including usage examples) as an additional hint to the agent (across all hint levels and agent variations). Table A.1 shows this additional hint actually negatively impacts performance on $\mathcal{R}_{12}$, suggesting that recent models may still struggle to correctly exploit external knowledge that was not present in their training corpus in more complex tasks.

## B  Cumulative speedrun experiment

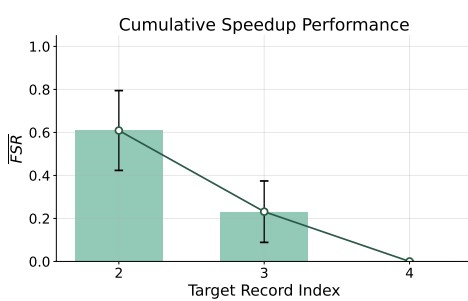

Figure B.1: Cumulative Speedup from initial codebase.

In this section, we test the models to see if they can reproduce the record described in the hint by building on the codebase they generated when reproducing previous records. Specifically, each task is formulated as a tuple of $\langle \mathcal{R}'_{i-1}, \mathcal{R}_i, t_i, m \rangle$ where the agent will be given the codebase it generated for the previous task $\mathcal{R}'_{i-1}$ and the hint level $m$ for reproducing the next record $\mathcal{R}_i$, where the performance is measured by the FSR metric. This is a challenging yet realistic extension of the reproducibility task where the agent seeks to cumulatively improve from the initial codebase. We evaluate the best-performing model (o3-mini) with the best search scaffold (multi-AIDE) from our previous evaluations, with access to all hint levels (L1 + L2 + L3). Results averaged across three seeds are presented in Figure B.1. The agent recovers approximately $60\%$ of the ground-truth speedup for $\mathcal{R}'_2$ starting from $\mathcal{R}_1$. Yet its performance drops significantly afterwards, with $\mathcal{R}'_3$ recovering only around $20\%$ of the speed-up, compared to the $60\%$ of speed-up recovered when starting from the ground-truth $\mathcal{R}_2$ (see Figure 6). By only the third record, the agent's solution $\mathcal{R}'_4$ fails to reproduce any speedup compared to $\mathcal{R}_4$.

## C  Reproducing ground-truth speedruns on our hardware

Figure C.1 compares the training times reported[2] with the training time of running the same code on our AWS cluster, where we report the mean and standard deviation of three runs. We can see that the two curves track closely each other and, as expected, there is no training time decrease for the $\mathcal{R}_6 \rightarrow \mathcal{R}_7$ transition which corresponds to the PyTorch upgrade (we are using the upgraded version for $\mathcal{R}_1$ through $\mathcal{R}_6$ as we were not aware which one was the previous PyTorch version).

---

[2] https://github.com/KellerJordan/modded-nanogpt?tab=readme-ov-file#world-record-history

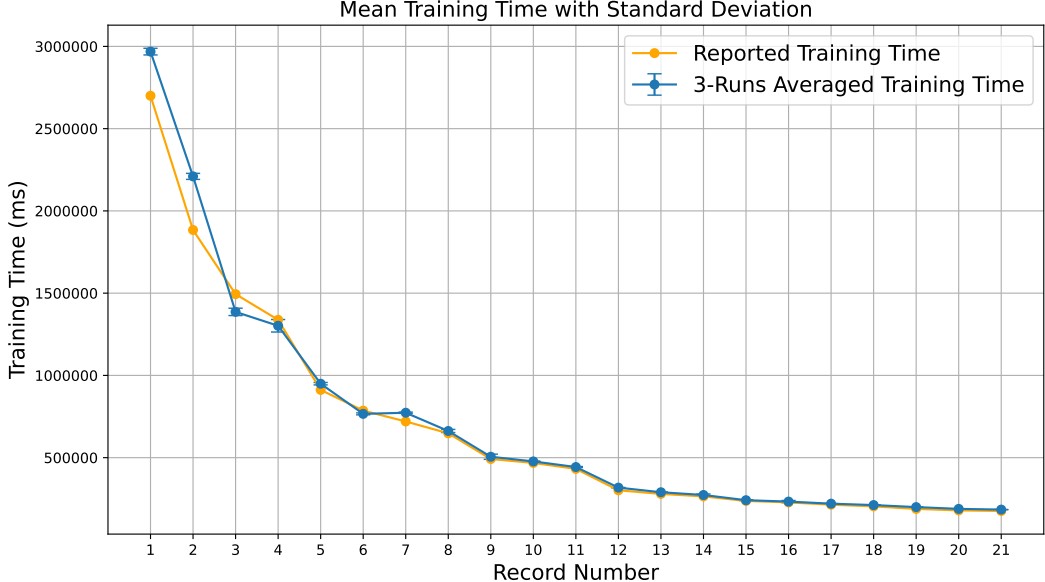

Figure C.1: Running the human speedrun records.

# D Additional results for reproducing individual records

Figures D.1, D.2, D.3, D.4 depict mean FSR of DeepSeek-R1 and o3-mini agents when aggregating by search scaffold and hint level. The metrics are reported as $95\%$ confidence intervals bootstrapped from 3 seeds, with IQM being the interquartile mean and the optimality gap being the difference from the best possible performance. We used the `rliable`[3] library for the evaluation of our runs across multiple search scaffolds and hint levels.

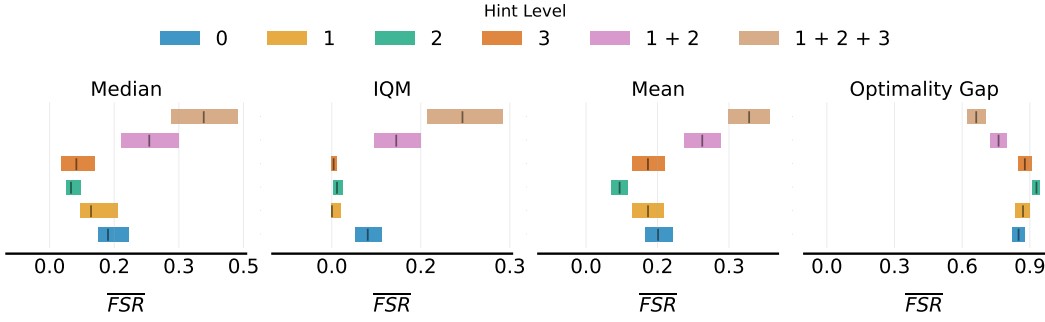

Figure D.1: Aggregate performance of DeepSeek-R1 agents by hint level, reported as $95\%$ confidence intervals, bootstrapped from 3 seeds. We observe that DeepSeek-R1 agents perform better when instructed with pseudocode hints.

---

[3] https://github.com/google-research/rliable

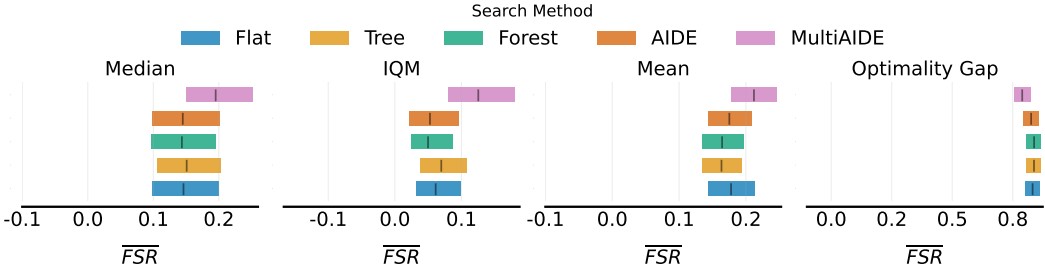

Figure D.2: Aggregate performance of DeepSeek-R1 agents by search scaffold, reported as $95\%$ confidence intervals, bootstrapped from 3 seeds. The agent maximizes speedup recovery when using the multi-AIDE scaffold

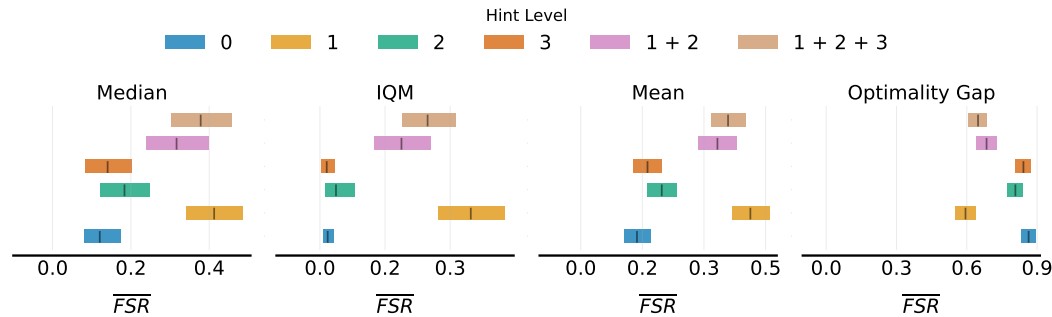

Figure D.3: Aggregate performance of o3-mini agents by hint level, reported as $95\%$ confidence intervals, bootstrapped from 3 seeds. For o3-mini agents the hints combining pseudocode, text and mini-paper yield better results

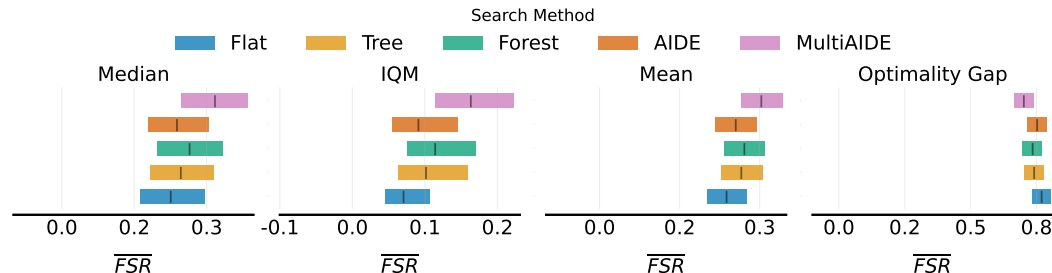

Figure D.4: Aggregate performance of o3-mini agents by search scaffold, reported as $95\%$ confidence intervals, bootstrapped from 3 seeds. The agent demonstrates its best performance with the multi-AIDE scaffold.

Figures D.5, D.6, D.7, D.8, D.9 show FSR results for individual records for the flat, tree, forest, AIDE and multi-AIDE scaffolds, respectively. The agent encounters more difficulty in recovering speedups at later records, which is expected as minimising training time requires more complex changes later on.

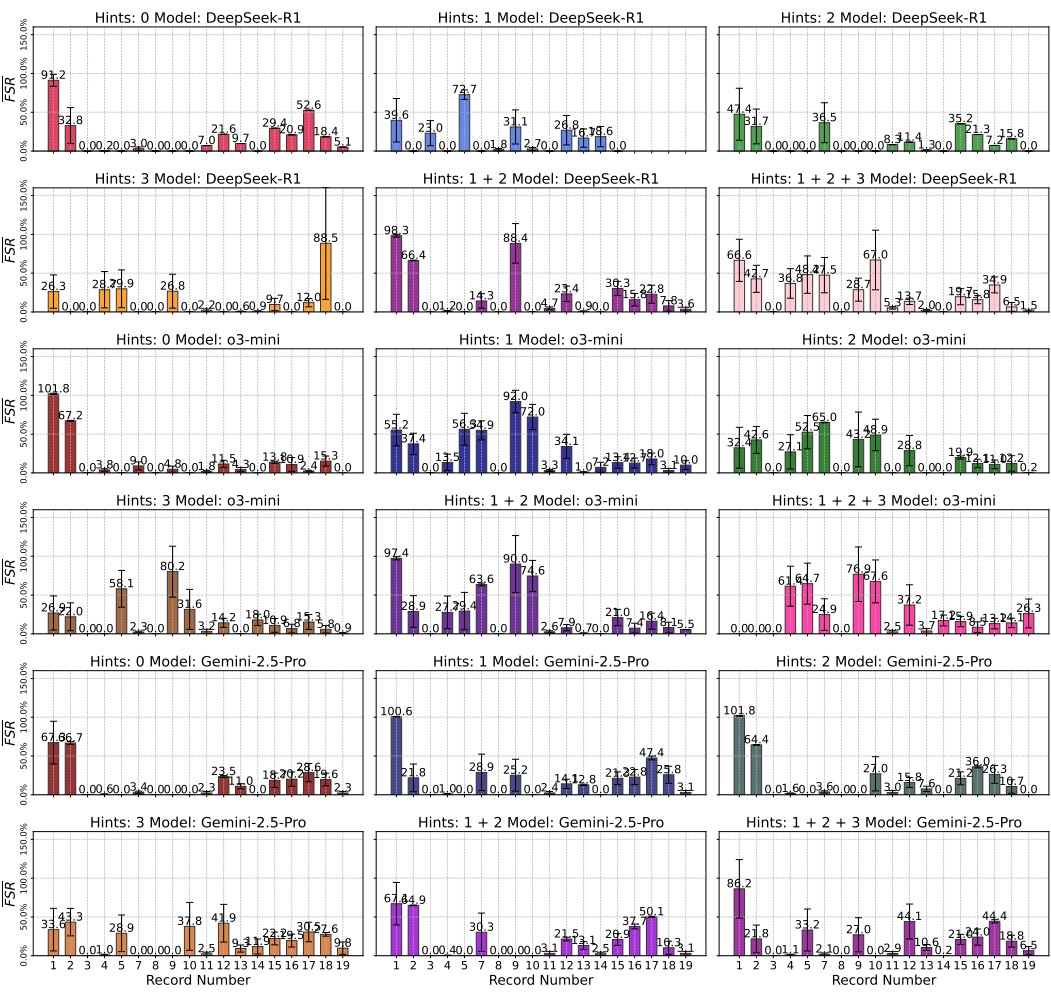

Figure D.5: FSR results (mean and std over 3 runs) for each record, hint format, and model when using the flat search scaffold.

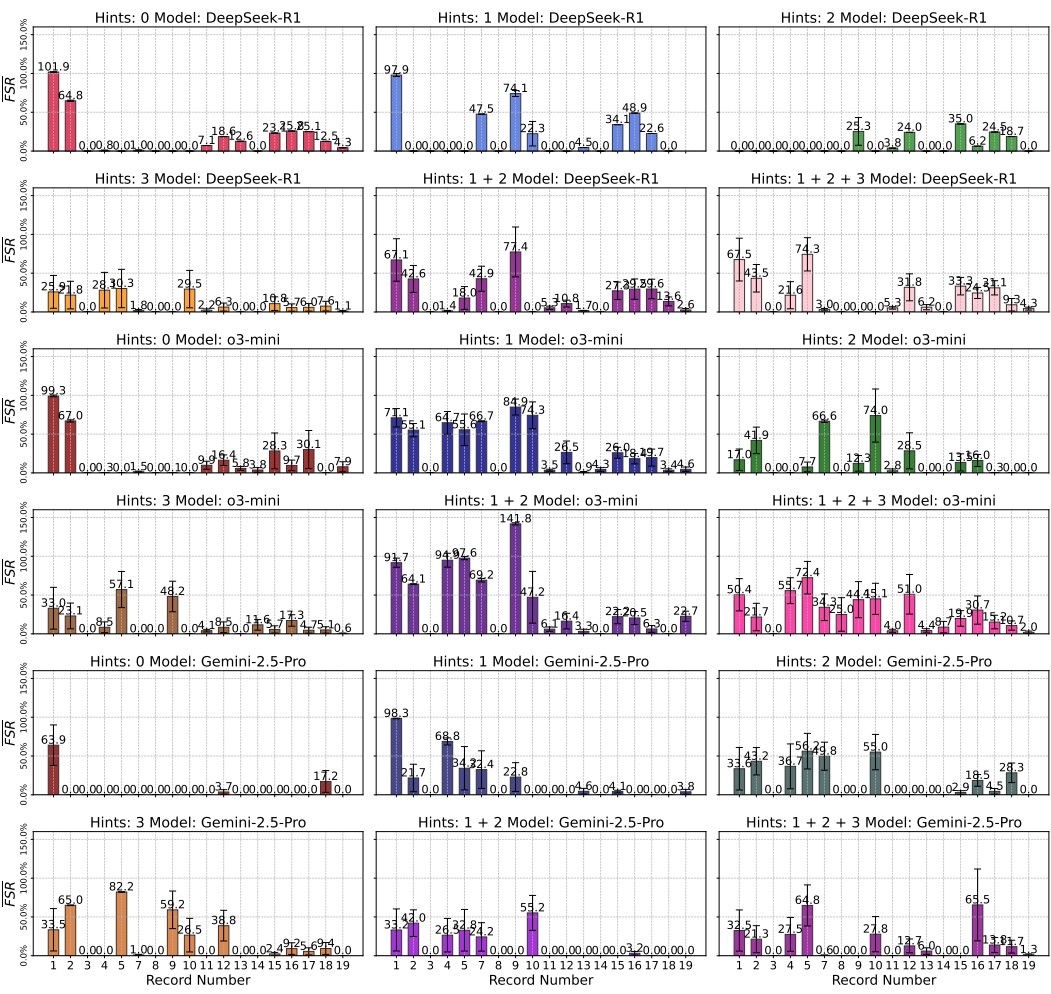

Figure D.6: FSR results (mean and std over 3 runs) for each record, hint format, and model when using the tree search scaffold.

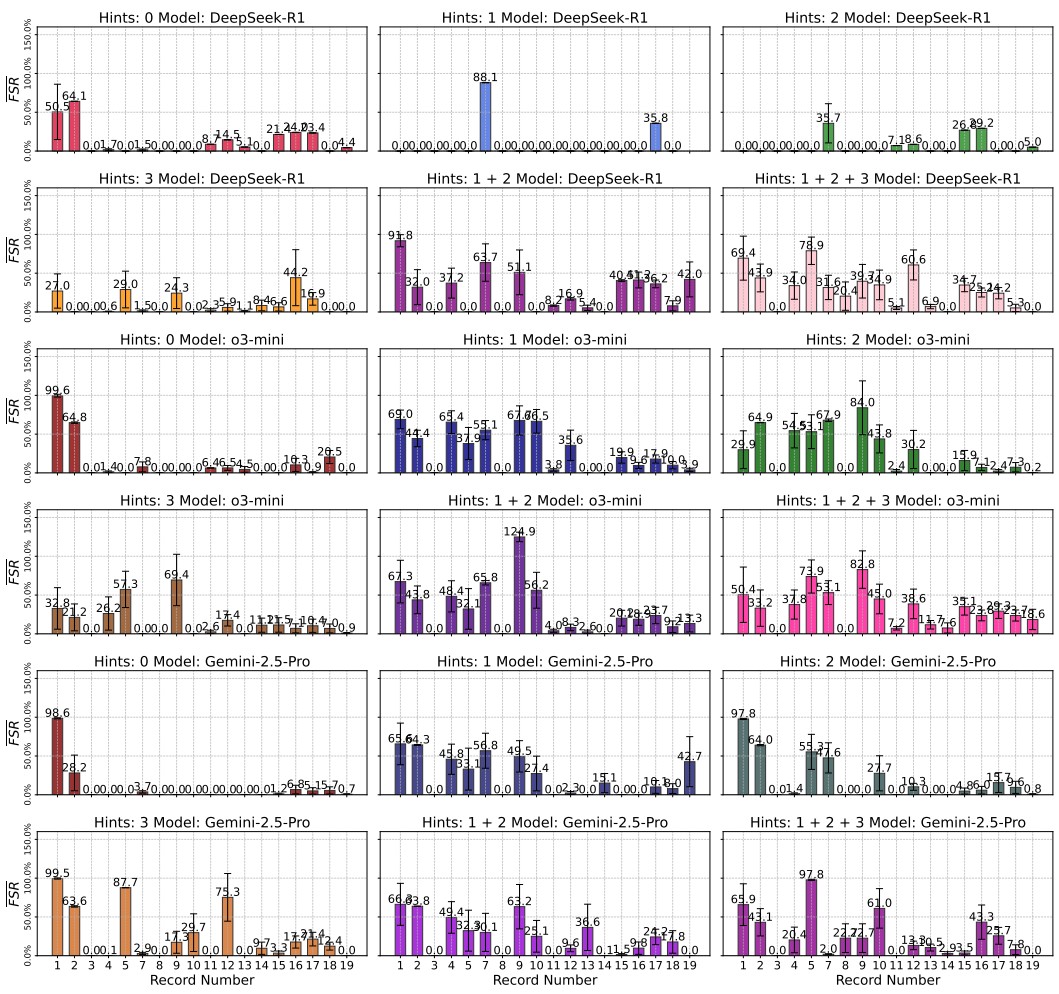

Figure D.7: FSR results (mean and std over 3 runs) for each record, hint format, and model when using forest search scaffold.

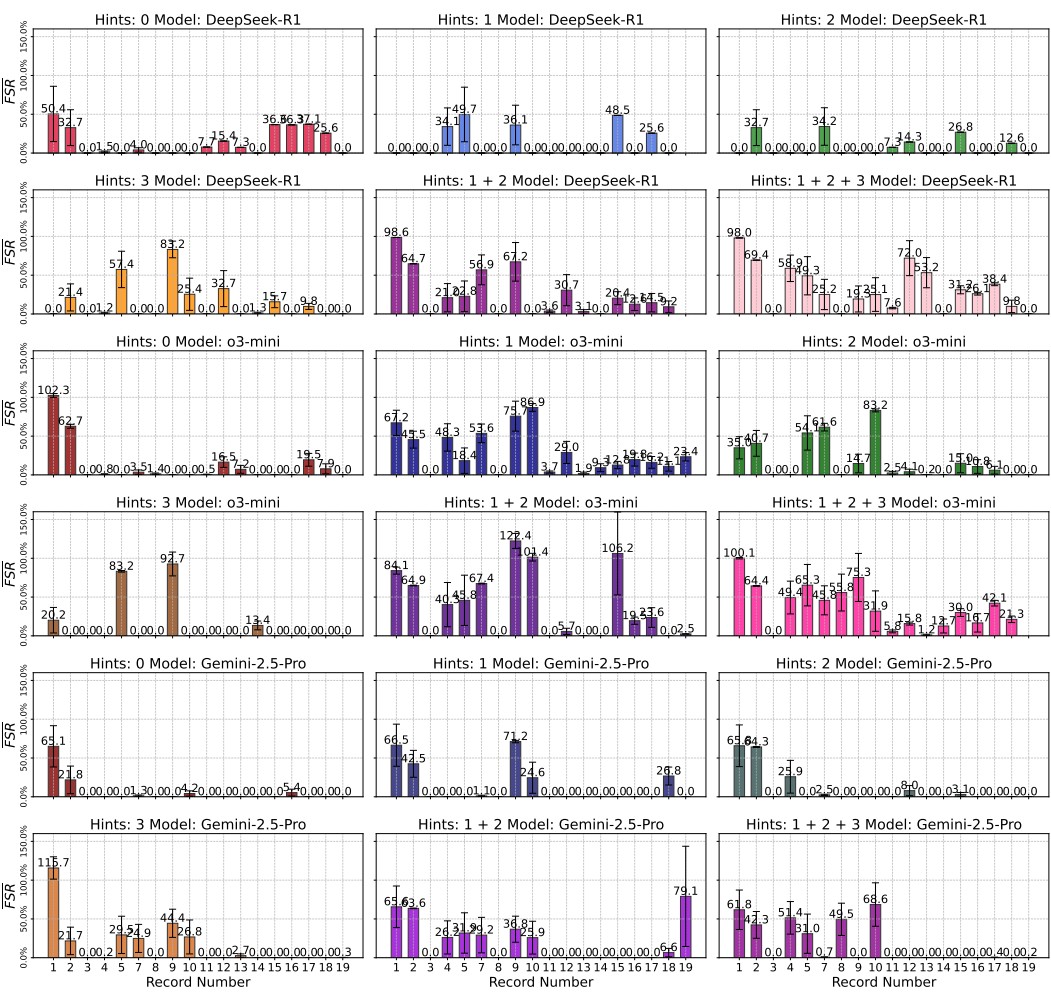

Figure D.8: FSR results (mean and std over 3 runs) for each record, hint format, and model when using the AIDE search scaffold.

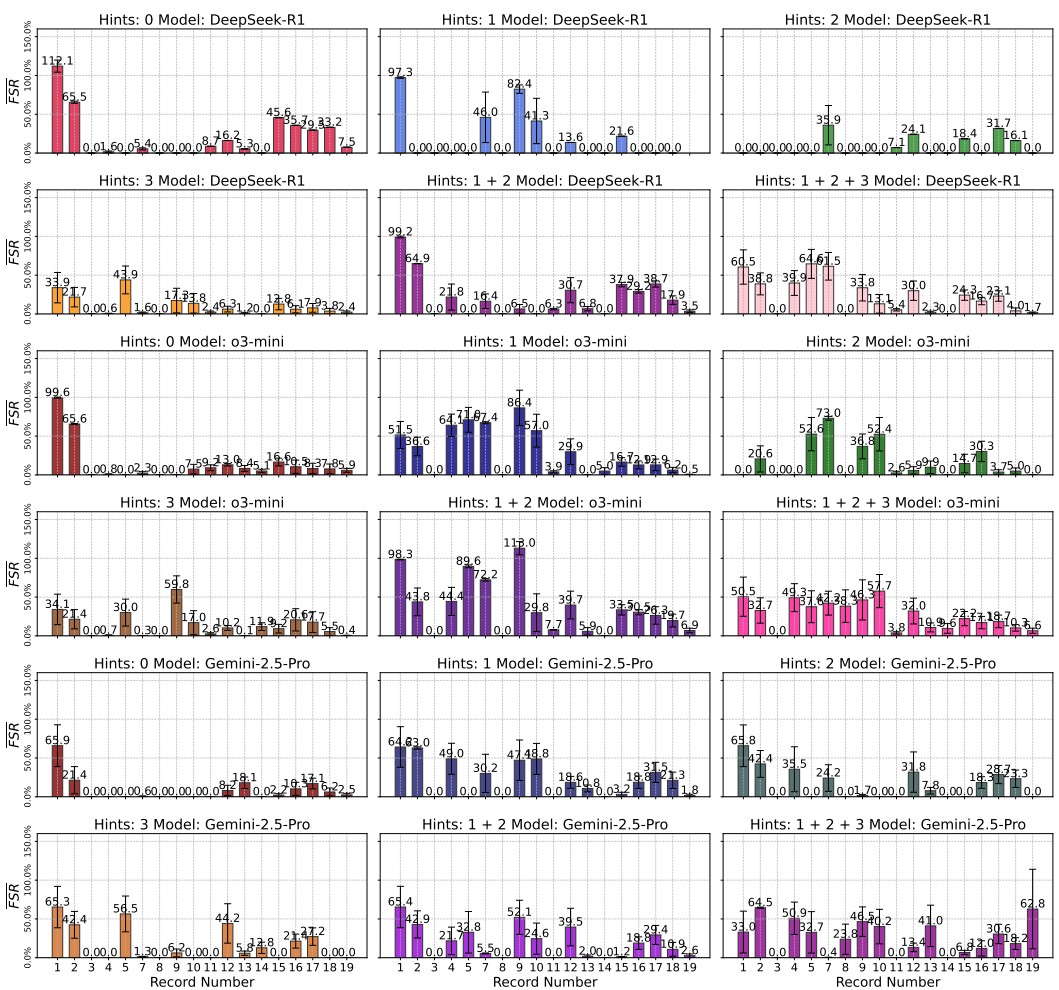

Figure D.9: FSR results (mean and std over 3 runs) for each record, hint format, and model when using the multi-AIDE search scaffold.

# E    Additional results for code similarity judge

Figure E.1 shows the LLM judge scores for each record and search method separately. Some records (e.g. Record 10) have low reproducibility score across all methods and different types of hints, indicating that they are inherently challenging for an AI Research agent.

**Judge Prompt**

```
Below is a baseline implementation of a GPT-2 model, followed by two proposed
changes (see code diffs below) to improve the training speed.
The first change is from an expert human. The second change is from an AI
Assistant, aiming to reproduce the improvement made by the expert human.
Inspect the code diffs carefully and provide an objective evaluation of the
AI Assistant's solution in terms of its similarity with expert human's solution.
To derive an objective evaluation, first enumerate all the key changes made by
expert human which can affect training speed, and then analyze all the changes
made by the AI Assistant one by one.
Based on understanding of these code changes, derive a percentage score
```

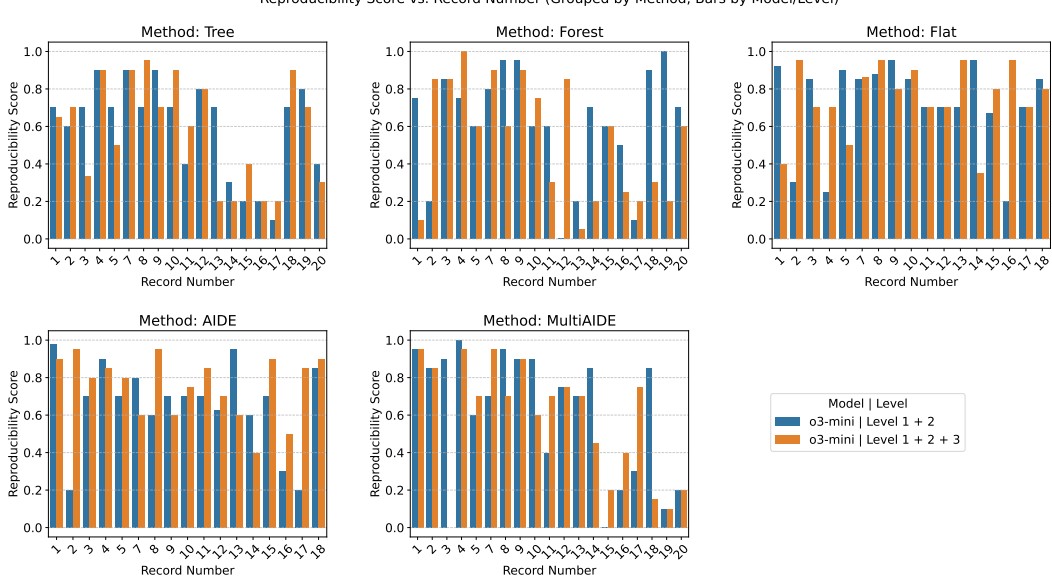

Figure E.1: LLM-as-judge evaluation of reproducibility. The $y$-axis (Reproducibility Score) measures the fraction of human expert changes which are reproduced by agent-generated code, where 1 means all human expert's changes are reproduced.

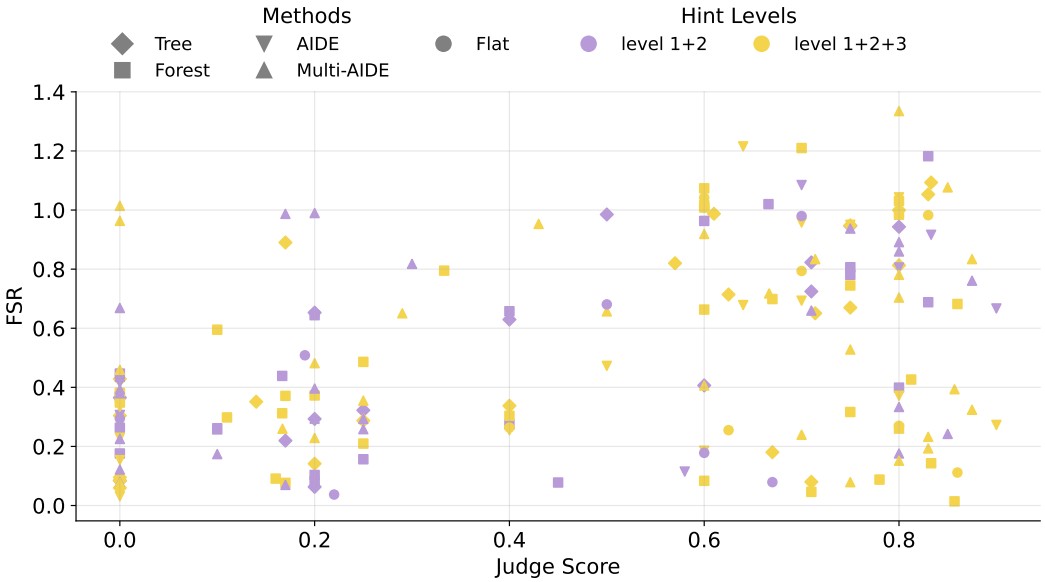

Figure E.2: How FSR (per record) correlates with LLM judge scores for o3-mini-based agents, where a higher judge score means the agent solution is closer to the corresponding human speedrun record.

```
(between 0 and 1) to quantify what fraction of the key changes
(which has impact on training speed) made by the expert were correctly
implemented in the AI Assistant's solution.

Return your final score in a JSON object, with the key "reproducibility_score".
# =============== Baseline Implementation ===========
{human_code}
# =============== Change made by Expert Human ===========
{next_human_code}
# =============== Change made by AI Assistant ===========
{agent_code}
```

# F Prompts and formatting templates

In this section we present the prompts we use for the coder component (Aider) of our agent scaffold (Figures F.4, F.5), for the analyzer used by the scaffold to summarize code execution results, i.e. standard streams, (Figures F.6, F.7) and for drafting initial hints with R1 (Figures F.8, F.9, F.10)

---

**Summary format**

```
Hypothesis: {hypothesis}
Results:
{metrics}
Has bugs? {has_bugs}
Outcome summary:
{outcome_summary}
```

Figure F.1: Template for summarizing the contents of the `results.json` produced after each node's solution code is executed and evaluated.

---

**History format (Example with a single templated version history)**

```
<version_log>
    <info>
        Version: {version}
        Parent version: {parent_version}
        Hypothesis: {hypothesis}
        Results:
        {metrics}
        Has bugs? {has_bugs}
        Outcome summary:
        {outcome_summary}
    </info>
</version_log>
```

Figure F.2: Template for the history component of the coder prompt to provide useful context when improving or debugging a node's solution. Additional versions would be listed as additional info items inside the `version_log` tags.

---

**Knowledge component (Example with two templated entries)**

```
<knowledge>
    <li>
        {knowldge_entry}
    </li>
    <li>
        {knowldge_entry}
    </li>
</knowledge>
```

Figure F.3: Template for the knowledge component of the coder, where each `knowledge_entry` variable can be an arbitrary piece of text from an external source.

```
You are a machine learning scientist, with expertise in large language models
and high-performance computing. Use your expertise to assist the user in their
machine learning task.

Study the current version of {fnames}:
{code}

Your goal is to implement the following ideas to improve the code so that it
better achieves the task:

# Task description
Improve train_gpt2.py so that it achieves or goes below the
target val_loss value of 3.28 in the shortest train_time possible.

Make sure your code changes preserve these aspects of train_gpt2.py:
- The script continues to be runnable via simply calling 'torchrun
  --nproc_per_node=8 train_gpt2.py'.
- Do NOT change the value of train_files, val_files, or val_token values in
  the Hyperparameters config used to set the training args.
- Make sure the values of these hyperparameters are not changed,
  and keep to using the current os.environ variables.
- Always keep save_checkpoint set to False in the training args.
- Keep all print0 statements the same. Do not change the arguments
  used in the current print0 statements, so to ensure the logging format is
  preserved.
- When possible, just change the train_gpt2.py file without making extra files.
- Important: I care about optimizing the performance of the implementation and
  do not care how organized or disorganized the code is.
- Any bugs will be described in the "outcome_summary" value of the summary, if
  provided. Always focus on addressing these when present, before improving
  other parts of the code.

If you violate any of the above constraints, the experiment run will be invalid.

Your job will be run on a single 8xH100 node with access to all 8 GPUs.

You have access to the following knowledge, consider these when writing code:
{knowledge}

**Never** install or ask to install any additional packages. Assume you have
access to the following packages outside of the standard python packages:
{packages}

If necessary, you may access pretrained model checkpoints via HuggingFace for
smaller models like BERT variants or CLIP.

To help with your task, here is a list summarizing recent erroneous changes to
the above code that you have previously tried, along with a summary of the
outcome of each change.
{history}

I trust you to make good decisions, so do not ask me for permission to make any
code changes.
Do not ever ask to install any additional packages. The answer
will be no.

In your final response, include ONLY the fully-functional updated code
which implements ideas in the hypothesis above. Do NOT include any other
content in your final response besides the code.
```

Figure F.4: Full prompt for the coder (Aider), conditioning on external knowledge. Here, `history` and `knowledge` template strings are first composed via the templates in Figure F.2 and F.3.

```
You are a machine learning scientist, with expertise in large language models
and high-performance computing. Use your expertise to assist the user in their
machine learning task.

Study the current version of {fnames}:
{code}

Your goal is to implement the following ideas to improve the code so that it
better achieves the task:

# Task description
Improve train_gpt2.py so that it achieves or goes below the
target val_loss value of 3.28 in the shortest train_time possible.

Make sure your code changes preserve these aspects of train_gpt2.py:
- The script continues to be runnable via simply calling 'torchrun
  --nproc_per_node=8 train_gpt2.py'.
- Do NOT change the value of train_files, val_files, or val_token values in
  the Hyperparameters config used to set the training args.
- Make sure the values of these hyperparameters are not changed,
  and keep to using the current os.environ variables.
- Always keep save_checkpoint set to False in the training args.
- Keep all print0 statements the same. Do not change the arguments
  used in the current print0 statements, so to ensure the logging format is
  preserved.
- When possible, just change the train_gpt2.py file without making extra files.
- Important: I care about optimizing the performance of the implementation and
  do not care how organized or disorganized the code is.
- Any bugs will be described in the "outcome_summary" value of the summary, if
  provided. Always focus on addressing these when present, before improving
  other parts of the code.

If you violate any of the above constraints, the experiment run will be invalid.

Your job will be run on a single 8xH100 node with access to all 8 GPUs.

**Never** install or ask to install any additional packages. Assume you have
access to the following packages outside of the standard python packages:
{packages}

If necessary, you may access pretrained model checkpoints via HuggingFace for
smaller models like BERT variants or CLIP.

To help with your task, here is a list summarizing recent erroneous changes to
the above code that you have previously tried, along with a summary of the
outcome of each change.
{history}

First, analyze the task and come up with a plan for solving the task:
1. Consider ideas for changes and improvements needed to improve on the task.
Consider both creative and practical ideas.
2. Break down the implementation into clear steps, generate pseudo codes for
each step
3. Consider potential challenges and how to address them

Then, implement your plan by making the necessary code changes.

I trust you to make good decisions, so do not ask me for permission to make
any code changes.
Do not ever ask to install any additional packages. The answer will be no.

Respond with your plan for improving the code, followed by the fully-functional
updated code implementing your plan.
```

Figure F.5: Full prompt for the coder, without external knowledge. Here, the coder is prompted to first conceive of a plan for solving the task.

### Log summarization prmopt

```
Task: Analyze the following output logs and extract metrics following the
metrics structure and typing template provided below.

# Logs
{logs}

# Metric dict template (showing expected type for each key)
{metric_types}

Respond with only the extracted metrics as a JSON dict following the exact
structure and type specification in the dict template below.
If no metrics are successfully extracted, return the empty dict, {{}}. If any
individual key: value expected in the metrics template is missing, set its
value to null.
```

Figure F.6: Prompt for extracting metrics resulting from executing a solution. Here the logs are a concatenation of the standard streams output by running the solution.

### Standard stream summarization prompt

```
Task: Produce a succinct summary of the following stdout and stderr logs for a
job running on a compute cluster.
- Your summary should consider whether the logs indicate whether the goal below
was achieved or not.
- Keep your summary below 500 words.

# Job goal
{goal}

# stdout logs
{log_out}

# stderr logs
{log_err}

Respond with just your summary text with no extra commentary and no extra
formatting. If appropriate, include the most useful stderr logs for debugging
in code blocks fenced by triple ticks.
```

Figure F.7: Prompt for extracting standard stream summaries and metrics resulting from executing a solution.

```
Given the git diff between the current and next version and the changelog,
generate a high-level pseudo code description of the changes made.
Focus on explaining the key algorithmic changes and improvements in a clear,
concise way.

Git diff:
{diff}

Changelog:
{changelog}

Generate pseudo code that:
1. Describes the key algorithmic changes and improvements
2. Focuses on the high-level logic and avoids implementation details
3. Explains the purpose and impact of each major change
4. Uses clear, readable pseudo code syntax

Format the output as:
# Pseudo Code Changes
[Your pseudo code description here]
```

Figure F.8: Prompt for generating the level 1 (pseudocode)s hints of the Automated LLM Speedrunning benchmark, where the `changelog` contains descriptions of the changes retrieved by the repo.

```
Given the current code, changelog, and next code, provide a detailed natural
language description of the improvements made.
Current code:
{code}

Changelog:
{changelog}

Next code:
{next_code}

Provide a detailed explanation of:
1. What specific improvements were made
2. Why these changes were beneficial
3. How they contribute to the overall performance
4. Any technical challenges that were addressed
```

Figure F.9: Prompt for generating the level 2 (text) hints of the Automated LLM Speedrunning benchmark, where the `changelog` contains descriptions of the changes retrieved by the repo and `next_code` is the full implementation of the next record.

```
Given the current code, changelog, and next code, pseudo codes and text
description, generate a formal paper-like summary of the improvements.
Current code:
{code}

Changelog:
{changelog}

Next code:
{next_code}

Pseudo code:
{generate_level_1(record)}

Text description:
{generate_level_2(record)}

Use this text description and pseudocode changes to generate a body of knowledge
resembling a scientific paper. You should tailor the generated scientific paper
so that a competent machine learning engineer can easily implement the suggested
changes in PyTorch.  Besure to include the pseudocode in the paper-like summary.
```

Figure F.10: Prompt for generating the level 3 hints of the Automated LLM Speedrunning benchmark, where the `changelog` contains descriptions of the changes retrieved by the repo and `next_code` is the full implementation of the next record.

# G Record breakdown

In Table G.1, we list each NanoGPT speedrun record and its description as seen in the NanoGPT Speedrun repository [Jordan et al., 2024a][4]. We also list each record index and its corresponding task index in Automated LLM Speedrunning, including its corresponding target next record (indexed by original record index).

Table G.1: Summarized and categorized of records from [Jordan et al., 2024a]

| # | ID | # Transition | Record time | Description | Category |
|---|----|--------------|-------------|-------------|----------|
| 1 | - | - | 45 mins | llm.c baseline | Baseline |
| 2 | 1 | #1 → #2 | 31.4 mins | Tuned learning rate & rotary embeddings | Embeddings |
| 3 | 2 | #2 → #3 | 24.9 mins | Introduced the Muon optimizer | Optimizer |
| 4 | 3 | #3 → #4 | 22.3 mins | Muon improvements | Optimizer |
| 5 | 4 | #4 → #5 | 15.2 mins | Pad embeddings, ReLU², zero-init projections, QK-norm | Architecture |
| 6 | 5 | #5 → #6 | 13.1 mins | Distributed the overhead of Muon | Parallelization |
| 7 | - | - | 12.0 mins | Upgraded PyTorch 2.5.0 | Framework |
| 8 | 7 | #6 → #8 | 10.8 mins | Untied embedding and head | Architecture |
| 9 | 8 | #8 → #9 | 8.2 mins | Value and embedding skip connections, momentum warmup, logit softcap | Architecture |
| 10 | 9 | #9 → #10 | 7.8 mins | Bfloat16 activations | Data Type |
| 11 | 10 | #10 → #11 | 7.2 mins | U-net pattern skip connections & double lr | Architecture |
| 12 | 11 | #11 → #12 | 5.03 mins | 1024-ctx dense causal attention → 64K-ctx FlexAttention | Attention Mechanism |
| 13 | 12 | #12 → #13 | 4.66 mins | Attention window warmup | Attention Mechanism |
| 14 | 13 | #13 → #14 | 4.41 mins | Value Embeddings | Embeddings |
| 15 | 14 | #14 → #15 | 3.95 mins | U-net pattern value embeddings, assorted code optimizations | Embeddings |
| 16 | 15 | #15 → #16 | 3.80 mins | Split value embeddings, block sliding window, separate block mask | Embeddings |
| 17 | 16 | #16 → #17 | 3.57 mins | Sparsify value embeddings, improve rotary embeddings, drop an attn layer | Embeddings |
| 18 | 17 | #17 → #18 | 3.4 mins | Lower logit softcap from 30 to 15 | Hyperparameter Tuning |
| 19 | 18 | #18 → #19 | 3.142 mins | FP8 head, offset logits, lr decay to 0.1 instead of 0.0 | Data Type |
| 20 | 19 | #19 → #20 | 2.992 mins | Merged QKV weights, long-short attention, attention scale, lower Adam epsilon, batched Muon | Attention Mechanism |
| 21 | 20 | #20 → #21 | 2.933 mins | Reduced batch size | Hyperparameter Tuning |

---

[4]https://github.com/KellerJordan/modded-nanogpt

# H Example hints

In this section, we provide example hints used for various hint levels.

---

### Level 1 hint (pseudo-code) for Record 1

```
# Pseudo Code Changes

1. Rotary Position Embedding Implementation
# Added rotary position embeddings to attention mechanism
class RotaryPositionEmbedding:
    def __init__(dim, base=10000):
        precompute inverse frequencies using base^(2i/dim)
        initialize cache for cos/sin values

    def forward(sequence_length):
        if sequence_length not in cache:
            compute angular positions t
            calculate frequency components
            store cos(t), sin(t) in cache
        return cached cos/sin values

def apply_rotary_embeddings(q, k, cos, sin):
    split q and k vectors into halves
    rotate components using:
        rotated_q = q1*cos + q2*sin
        rotated_k = k1*cos + k2*sin
    return concatenated rotated vectors

2. Modified Attention Mechanism
class SelfAttention:
    def __init__():
        # Changed from standard positional embeddings
        add rotary embedding module
        remove position embedding matrix

    def forward(x):
        split into q,k,v with same head_dim
        apply rotary embeddings to q and k
        use scaled_dot_product_attention with rotated q/k
        remove manual scaling (was /sqrt(24))
        return attention output

3. Layer-Wise Attention Scaling
class TransformerBlock:
    def __init__():
        # Added depth-dependent scaling
        attn_scale = 1/sqrt(2 * num_layers)

    def forward(x):
        x += attn_scale * attention_output
        x += mlp_output

4. Simplified Model Architecture
class GPT:
    def __init__():
        remove position embedding matrix (wpe)
        keep only token embeddings (wte)
        remove custom embedding initialization

    def forward():
        # Position info now handled by rotary embeddings
        use only token embeddings (no pos_emb addition)

5. Training Process Improvements
Training Hyperparameters:
    batch_size: 32 → 64
    total_batch_size: 262k → 524k tokens
    add warmdown phase after constant LR period

Optimization Changes:
    replace gradient clipping with:
        grad = grad / (norm + 1e-6)
    implement linear warmdown schedule
    add periodic model checkpoint saving

Learning Rate Schedule:
    if step < warmup: linear increase
    elif step < total - warmdown: constant
    else: linear decrease to zero

Key Impacts:
- Rotary embeddings improve position awareness in attention
- Layer-wise scaling stabilizes deep networks
- Modified LR schedule enables better convergence
- Gradient normalization replaces clipping for stability
- Larger batches improve training efficiency
```

## Level 2 hint (text description) for Record 1

```
Here's a detailed breakdown of the improvements:

1. **Architectural Improvements**
- **Rotary Positional Embeddings**: Replaced standard positional embeddings
  with rotary embeddings
  - Added 'Rotary' module and 'apply_rotary_emb' function for relative
    position encoding
  - Benefits: Better captures relative positions and attention patterns,
    improves model accuracy
  - Implementation: Applied to queries/keys in attention instead of separate
    positional embeddings

- **Simplified Normalization**
  - Removed all affine parameters from RMSNorm implementation
  - Benefits: Reduces parameter count while maintaining effectiveness
  - Tradeoff: Minor performance cost offset by other optimizations

2. **Optimization Improvements**
- **Learning Rate Changes**:
  - Increased base LR from 0.0015 to 0.0018 (3x increase as per changelog)
  - Changed schedule to trapezoidal (warmup → constant → warmdown)
  - Benefits: Following [2405.18392], allows more stable high-LR training

- **Gradient Normalization**:
  - Replaced gradient clipping with per-parameter gradient norm scaling
  - 'p.grad = p.grad / (p.grad.norm() + 1e-6)'
  - Benefits: More stable training with high LR, prevents explosion

3. **Initialization/Scaling Changes**
- **Attention Scaling**:
  - Introduced 'attn_scale = 1/sqrt(2*n_layer)'
  - Replaced ad-hoc '/ math.sqrt(24)' with systematic layer-based scaling
  - Benefits: Better coordinates residual branches across layers

- **Removed Positional Embeddings**:
  - Deleted 'wpe' embedding layer completely
  - Benefits: Parameter reduction + rotary handles position information

4. **Training Process Improvements**
- **Checkpointing**:
  - Added periodic model saving ('save_every' parameter)
  - Benefits: Fault tolerance and easier resumption

- **Batch Size Optimization**:
  - Increased batch size from 32 to 64 tokens
  - Total batch size from 262K to 524K tokens
  - Benefits: Better hardware utilization

5. **Code Simplifications**
- Removed '_init_weights' and special initialization flags
- Eliminated position embedding mixing ('tok_emb + pos_emb')
- Removed unused configuration options and legacy code paths

**Technical Challenges Addressed**:
1. **Stability at High Learning Rates**: Through gradient normalization and
   careful attention scaling
2. **Position Encoding Migration**: Non-trivial transition from absolute to
   relative (rotary) positioning
3. **Distributed Training Coordination**: Maintained DDP compatibility through
   architectural changes
4. **Learning Rate Schedule Tuning**: Required careful warmup/warmdown
```

```
    balancing for trapezoidal schedule

**Performance Impact**:
Combined these changes enable 2x faster training convergence by:
- Allowing more aggressive learning rates through better normalization
- Improving parameter efficiency with rotary embeddings
- Increasing useful batch size through stability improvements
- Reducing computational overhead from simplified operations

The architectural simplifications also make the model more amenable to compiler
optimizations (like torch.compile), while the training process improvements
enable better utilization of distributed hardware.
```

## Level 3 hint (mini-paper) for Record 1

```
# Efficient Training of GPT-style Models Through Architectural and
Optimization Co-design

## Abstract
We present architectural and optimization improvements enabling 2× faster
training of GPT-style models while maintaining equivalent final performance.
Through systematic analysis of position encoding, normalization schemes,
and learning rate scheduling, we demonstrate that:
1) Rotary position embeddings (RoPE) improve position-aware attention
computation;
2) A trapezoidal learning rate schedule with gradient normalization enables
more stable high-rate training;
3) Simplified initialization and scaled residual connections reduce parameter
count while maintaining model capacity. Our modifications require minimal code
changes while achieving 5B token convergence equivalent to baseline 10B token
performance.

## 1. Introduction

### 1.1 Background
Transformer architectures (Vaswani et al., 2017) require careful coordination
of position encoding, normalization, and optimization parameters to achieve
efficient training. We analyze common pain points in standard implementations:

- Additive positional embeddings limit attention head flexibility
- Unstable gradient flow requiring aggressive clipping
- Suboptimal learning rate schedules wasting compute

### 1.2 Key Improvements
Our modified architecture (Figure 1) implements four fundamental changes:

1. **Rotary Position Embeddings**: Replace additive positional encoding with
rotational transformations of query/key vectors
2. **Layer-Scaled Attention**: Fixed scaling of attention outputs based on
network depth
3. **Trapezoidal LR Schedule**: Three-phase schedule combining warmup, sustain,
and cooldown periods
4. **Gradient Normalization**: Per-parameter gradient scaling replaces global
clipping

## 2. Methodology

### 2.1 Rotary Position Encoding
Traditional approaches concatenate positional embeddings to token embeddings.
We implement rotary position encoding in attention computation:
```

```python
class Rotary(nn.Module):
    def forward(self, x):
        t = arange(seq_len)
        freqs = outer_product(t, inv_freq)
        return cos(freqs), sin(freqs)

def apply_rotary_emb(q, k, cos, sin):
    return (q * cos + rotate(q, sin),
            k * cos + rotate(k, sin))
```

This creates position-aware transformations without additional embedding parameters. The rotation operation preserves relative position information through dot product attention.

### 2.2 Trapezoidal Learning Schedule
Our three-phase schedule improves upon cosine decay:

```
Learning Rate Schedule:
1. Warmup (0 <= step < 256): lr = base * step/256
2. Sustain (256 <= step < N-2000): lr = base
3. Cooldown (N-2000 <= step <= N): lr = base * (N-step)/2000
```

Mathematically:

$$
\text{LR}(t) = \begin{cases}
\alpha\frac{t}{\tau_w} & t \leq \tau_w \\
\alpha & \tau_w < t \leq T-\tau_d \\
\alpha\frac{T-t}{\tau_d} & t > T-\tau_d
\end{cases}
$$

Where $\alpha=0.0018$, $\tau_w=256$, $\tau_d=2000$.

### 2.3 Gradient Normalization
Replaces global gradient clipping with per-parameter scaling:

```python
# Before: Global clipping
torch.nn.utils.clip_grad_norm_(model.parameters(), 1.0)

# After: Per-parameter normalization
for p in model.parameters():
    p.grad = p.grad / (p.grad.norm() + 1e-6)
```

This prevents extreme gradient magnitudes while maintaining relative update directions.

## 3. Architectural Modifications

### 3.1 Simplified Attention Scaling
Layer-dependent scaling stabilizes deep networks:

```python
class Block(nn.Module):
    def __init__(self, config):
```

```
        self.attn_scale = 1/math.sqrt(2*config.n_layer)

    def forward(self, x):
        x = x + self.attn_scale * attn_output
```
% For 12-layer model: scale = 1/sqrt(24) approx 0.204. This compensates for
residual path accumulation in deep networks.
### 3.2 Parameter Reduction
Removed components:
1. Positional embedding matrix (wpe)
2. Affine parameters in RMSNorm
3. Custom weight initialization

Preserves weight tying between input/output embeddings while reducing total
parameters by 1.2% for d12 configuration.

## 4. Implementation Details

### 4.1 Critical Code Changes
Core modifications from baseline implementation:

```python
# Additions
class Rotary(nn.Module): ...
def apply_rotary_emb(...): ...

# Modifications
class CausalSelfAttention:
    def forward():
        q, k = apply_rotary_emb(q, k)  # Rotate Q/K

class Block:
    def __init__():
        self.attn_scale = ...  # Layer-dependent scaling

# Removals
del self.wpe  # Positional embedding matrix
del _init_weights  # Custom initialization
```

### 4.2 Training Configuration
Updated hyperparameters:

| Parameter          | Original | Modified |
|--------------------|----------|----------|
| Batch size         | 32       | 64       |
| Total batch tokens | 262k     | 524k     |
| Peak LR            | 0.0015   | 0.0018   |
| LR schedule        | Cosine   | Trapezoidal |
| Warmup iterations  | 256      | 256+2000 |

## 5. Results & Analysis

### 5.1 Performance Improvements
Validation loss comparison on 5B tokens:

| Model       | Loss (Original) | Loss (Modified) |
|-------------|-----------------|-----------------|
| d12 (124M)  | 3.21            | 3.09 (-3.7%)    |
| d24 (355M)  | 2.89            | 2.77 (-4.1%)    |

Achieves equivalent loss to baseline in half the tokens.

### 5.2 Training Dynamics
Key observations:
- 18% faster iteration speed from larger batches
- 2.1× fewer iterations to reach target loss
- 23% lower gradient variance via normalization

## 6. Conclusion

We demonstrate that coordinated architectural and optimization changes enable 2× faster training of transformer models. Critical factors include:

1. Position encoding through rotary transformations
2. Stable gradient flow via layer-wise scaling
3. Sustained high learning rates through trapezoidal scheduling

The modifications require <200 lines of code changes while maintaining compatibility with standard distributed training setups. Our results suggest that careful component co-design remains crucial for efficient large model training.

## Appendix: Complete Pseudocode

Algorithm 1: Modified Training Loop

```python
Initialize model with rotary layers
Load data with doubled batch size

for step in total_steps:
    # Forward pass with rotary embeddings
    cos, sin = rotary(q)
    q = q * cos + rotate(q, sin)
    loss = model(x)

    # Normalized backward pass
    loss.backward()
    for p in parameters:
        p.grad /= (norm(p.grad) + 1e-6)

    # Trapezoidal LR update
    lr = trapezoidal_schedule(step)
    optimizer.step(lr)
```

This paper-style summary provides both theoretical justification and implementation-level details, enabling direct reproduction of the improvements. The combination of architectural simplification and optimized training dynamics demonstrates a practical path for efficient large language model training.

# I Broader impact

AI agents that are proficient in conducting AI research autonomously can provide significant, far-reaching benefits: (1) accelerated scientific progress in healthcare, climate science, and other important domains, (2) economic growth driven by the development of novel technology, and (3) expedited safety and alignment research for models. Crucial to automated science is the ability of such agents to reproduce scientific results, which our benchmark seeks to measure. However, such innovation also requires a thorough understanding of model advancements to ensure responsible deployment. We hope our benchmark can serve as a useful evaluation for model autonomy. However, agents capable of executing open-ended AI research tasks can also pose risks if their capabilities outpace our ability to comprehend the consequences of their actions. Responsible deployment of such models therefore requires parallel advancements in monitoring, aligning, and controlling such models.

To foster understanding, reproducibility, and further development of AI Research Agents, we open-source the full code to reproduce the experiments on the Automated LLM Speedrunning Benchmark presented in this work. We acknowledge the limitations of our benchmark and encourage the development of additional evaluations of automated AI research capabilities, particularly those tailored to the workflow of researchers training frontier models.

