# OpenReview forum: "The Automated LLM Speedrunning Benchmark: Reproducing NanoGPT Improvements"
_NeurIPS.cc/2025/Datasets_and_Benchmarks_Track — NeurIPS 2025 Datasets and Benchmarks Track poster_

### Official Review · Reviewer_BaAM · 2025-06-24

**Rating:** 4
**Confidence:** 2

**Summary:**

This paper introduces the ​Automated LLM Speedrunning Benchmark for evaluating AI research agents’ ability to reproduce scientific improvements in code. Built on the NanoGPT Speedrun community effort, the benchmark requires agents to sequentially reimplement ​code-level optimizations​ between consecutive speedrun records. Success is measured by whether the agent’s modified script ​reproduces the original record’s training results and speed. The authors find that even SOTA LLM agents with advanced scaffolds struggle to reproduce known optimizations despite detailed hints. This suggests ​automated scientific reproducibility remains a major bottleneck​ for autonomous research agents.

**Dataset Code Accessibility:**

Yes

**Ethical Considerations:**

No, there are no or only very minor ethics concerns

**Final Justification:**

The rebuttal have well addressed my concerns and I recommend for acceptance.

**Limitations Weaknesses:**

1. While optimizing GPT-2 training is valuable, it’s unclear if struggles here generalize to ​automated reproducibility to other models or tasks scenarios.

2. How do these tasks compare to traditional coding benchmarks? What makes them uniquely hard? High-level discussion would help grasp the challenge.

**Strengths Contributions:**

1. The key idea for building this benchmark is interesting and novel. Turning community-driven optimizations (NanoGPT Speedrun) into a structured benchmark for cumulative scientific reproduction is interesting. It measures a critical real-world skill for translating descriptions of incremental improvements into functional code.

2. The focus on ​automated reproducibility​ is well-argued and addresses a fundamental challenge in trustworthy AI-driven research.

3. The result that SOTA agents fail to reproduce known improvements even with hints is striking, underlining the benchmark’s utility and the difficulty of automated reproducibility.

---

> ### Author Rebuttal · Authors · 2025-07-29
>
> We thank Reviewer BaAM for their supportive review and for highlighting the novelty of the benchmark's core idea.
>
> **[BaAM-1] Do the issues observed for automated reproducibility on your benchmark generalize to other models or tasks scenarios?**
>
>
> We appreciate this question, as the generalizability of our benchmark's challenges is one of its key strengths. While the tasks are anchored in the NanoGPT speedrun, the scientific innovations within it have shown proven, broader impact.
>
>
> For instance, a core challenge is reproducing the Muon optimizer, which originated in this speedrun and has since been shown by other researchers to be effective for training much larger language models such as the Kimi K2 model. This real-world transfer demonstrates that our benchmark measures an agent's ability to reproduce discoveries with widespread relevance, making it a strong proxy for general scientific capability in the AI domain.
>
> While our benchmark's domain is AI research and does not cover wider scientific domains like biology or chemistry, it rigorously tests the universal scientific principle of reproducing an experiment from its description. This scientific principle is transferable to other domains. We appreciate the suggestion and will clarify this scope in the paper.
>
>
>
> **[BaAM-2] How do the tasks in your benchmark compare to traditional coding benchmarks?**
>
>
> This is an excellent question. Many standard coding benchmarks evaluate an agent's ability to implement well-known algorithms or solve self-contained logic problems. In contrast, our benchmark operates at the frontier of machine learning (ML) research. It requires agents to understand and implement novel and complex concepts such as the Muon optimizer or efficient attention variants like FlexAttention from high-level, paper-like descriptions. This involves a deeper understanding of ML engineering than is required in traditional coding benchmarks.
>
>
> Furthermore, our benchmark introduces several other layers of difficulty not typically found in standard coding tests:
> - **Focus on performance optimization:** The primary success metric is not just functional correctness but a measurable improvement in wall-clock training time, which is a much harder target.
> - **Modification of a complex codebase:** Agents must make precise modifications to an existing, non-trivial training script rather than writing code from scratch.
> - **Cumulative research trajectory:** The tasks can be made sequential, meaning success in later stages depends on the cumulative changes from all prior stages, mimicking a real research process.
> - **Range of abstractions:** The tasks in our benchmark are supplemented with different hint types that provide a range of abstractions of the solution in the context.

---

> > ### Comment · Reviewer_BaAM · 2025-08-05
> > **Thanks for the rebuttal**
> >
> > The author's rebuttal has effectively addressed my concern. Therefore, I'm happy to maintain my score.

---

### Official Review · Reviewer_JERV · 2025-06-29

**Rating:** 4
**Confidence:** 4

**Summary:**

NanoGPT is an internet-famous, community-driven effort to minimize FineWeb loss on an 8xH100 node using a PyTorch implementation of GPT-2. Due to its public nature and immense amount of public communication of progress, and incredibly detailed documentation of incremental improvements, it becomes a great candidate for the creation of scientific reproducibility challenges. This manuscript does just that - a benchmark that tries to assess reasoning LLM's capabilities to incrementally re-create the NanoGPT speedrun trajectory. The authors have created a codebase, and corresponding experiments on o3-mini and DeepSeekR1 to see if different variants of coding agents (e.g. AIDE) can advance through different iterations of the changelog given various forms of supervision (from pseudocode to mini paper) derived from the publicly available records. The authors have found out that existing, frontier coding agents struggle at replicating these records even with ample hints provided.

**Dataset Code Accessibility:**

Yes

**Dataset Code Comments:**

The code is readily available at https://drive.google.com/file/d/1edVdWtc3XJRRzcxFeWUazjc9vUzUHkVp/edit and upon downloading I'm able to view individual run results under `data`. The framework appears to be complete, and there are well-written documentations on how to reproduce the results.

**Ethical Considerations:**

No, there are no or only very minor ethics concerns

**Final Justification:**

Rebuttal clears up remaining minor concerns. Speedrunning is an exciting benchmark - I lean towards acceptance and maintain my score.

**Limitations Weaknesses:**

I have two main concerns with this work:
1. The set up is still slightly contrived and not general - the claim that reproducing NanoGPT serves as a generalizable benchmark for scientific reproducibility seems under-explained. Many scientific tasks do not have ample public record on iterative improvements, some define new tasks, other challenge existing formulations.
2. Admittedly the benchmark is undersaturated, but the fact that even the strongest reasoning model struggles with long-horizon, complex, and agentic tasks is expected as of now. I'd appreciate more insights on why this particular low-performance result is interesting.

These are minor concerns. I lean towards acceptance.

**Strengths Contributions:**

NanoGPT is a great public experiment to be based upon in regards for a reproducibility benchmark, and the authors have done a solid job. As far as I'm aware this work is the first of its kind in terms of benchmarking, though the authors could enhance this point in discussing the relationship of this to existing works. Figures and captions are very informative, and the code is fully available, and appears to be complete and informative. The remark on similarity of successful AI-generated code and human code is interesting and fresh.

---

> ### Author Rebuttal · Authors · 2025-07-29
>
> We thank Reviewer JERV for their feedback and for recognizing the novelty of using NanoGPT as a foundation for our benchmark.
>
>
> **[JERV-1] Not a general benchmark for scientific reproducibility.**
>
>
> While our benchmark is focused on the specific research path of the NanoGPT speedrun, we argue that this specificity is what allows for a rigorous test of skills that are broadly applicable to scientific advancement.
>
>
> Crucially, the scientific breakthroughs within our benchmark have demonstrated significant real-world generalizability. A prime example is the Muon optimizer, which was a key discovery from the speedrun that agents are tasked with reproducing in our benchmark. This very optimizer has since been adopted by other researchers and proven to be effective for training much larger, modern LLMs.  This outcome suggests that findings in NanoGPT can be transferred and scaled up to larger-scale, more complex setups, even at the 1T parameter scale as shown in the Kimi K2 work.
>
> We agree that our benchmark, while focused on AI research, may not generalize to other scientific fields like biology or chemistry. However, it is designed to test a core principle fundamental to all automated science: the ability to reproduce an experiment from its description. We thank you for the suggestion and will update the paper to clarify this scope.
>
>
>
> **[JERV-2] The low performance results are expected.**
>
>
> We would argue that the degree of underperformance we observe is actually quite surprising. For example, on the SWE-bench Verified, o3-mini can solve 49.3% of tasks as shown in the official openai o3-mini blog, whereas on our benchmark, the best results do not exceed 40% FSR.
>
> Our benchmark's tasks, particularly those that include hints, are arguably simpler in one key respect; i.e., the "solution" is explicitly described within the prompt's context. The agent does not need to infer a solution from a vague description of a failure; it simply needs to translate a direct specification (in some cases, as detailed as pseudocode) into a correct and performant implementation within an existing codebase.
>
> The fact that even strong reasoning models struggle so significantly under these conditions is therefore unexpected and revealing. It points to a critical gap between high-level reasoning and the meticulous detail-oriented work of implementing a specific optimization correctly. We argue that this makes our results more than just an expected data point; they are an important diagnostic of a specific and challenging failure mode for current agents, highlighting that automated reproducibility remains a significant hurdle.

---

### Official Review · Reviewer_8M92 · 2025-07-01

**Rating:** 5
**Confidence:** 3

**Summary:**

This paper introduces a new benchmark, the Automated LLM Speedrunning Benchmark, designed to evaluate LLM agents' ability to reproduce performance improvements in the NanoGPT speedrun competition. The benchmark focuses on replicating cumulative code-level changes between successive training records that reduce wall-clock training time to a target loss. Each task is accompanied by optional hints, and evaluation is conducted via normalized speedup recovery. The authors demonstrate that even recent LLMs  struggle to reproduce prior records, highlighting automated reproducibility as a key bottleneck in developing autonomous research agents.

**Dataset Code Accessibility:**

Yes

**Ethical Considerations:**

No, there are no or only very minor ethics concerns

**Final Justification:**

Authors solved my concerns

**Limitations Weaknesses:**

1. Though highly relevant to LLM optimization, the benchmark’s narrow scope (NanoGPT speedrun) may limit generalization to broader scientific reproduction tasks.

2. The degradation of DeepSeek-R1 with richer hints is surprising but not deeply explained. More insight into this phenomenon would benefit future scaffold design.

3. While FSR is a solid metric, it abstracts away how close the code is to the original. The authors do include some embedding and judge-based comparisons, but they remain underdeveloped.

**Strengths Contributions:**

1. Unlike previous benchmarks that assess one-off tasks, this benchmark uniquely evaluates an agent’s ability to reproduce cumulative research findings across a realistic trajectory.

2. The Fraction of Speedup Recovered is a simple yet effective quantitative metric, grounded in actual training time.

3. The paper includes several informative diagnostics, including L2 embedding similarity, scaffold tree behavior, and performance breakdowns across hint formats.

---

> ### Author Rebuttal · Authors · 2025-07-29
>
> We thank Reviewer 8M92 for the insightful feedback and for recognizing the strengths of our work, including its cumulative nature, the FSR metric, and the informative diagnostic analyses.
>
>
> **[8M92-1] How can this benchmark generalize to broader scientific reproduction tasks?**
>
> This is an important question. While our benchmark is intentionally focused on the challenging NanoGPT speedrun, the scientific innovations the speedrun contains have proven to be highly generalizable in the real world. For instance, a key task in our benchmark is the reproduction of the Muon optimizer, an algorithmic enhancement that was discovered during the speedrun. This very optimizer has since been shown by other researchers to be effective and scalable for training much larger, state-of-the-art LLMs such as Kimi K2.
>
>
> The fact that core innovations from the NanoGPT speedrun transfer to frontier models provides strong evidence that our benchmark is not an isolated exercise. It tests an agent's ability to reproduce discoveries that are of direct relevance to the broader field of LLM research and development. Therefore, success on our benchmark is a strong proxy for an agent's capability to perform meaningful, generalizable scientific reproduction in the AI domain.
>
> While highly representative of many tasks in AI research and coding, it is indeed fair to say that our benchmark does not necessarily generalize to all tasks in other scientific disciplines, e.g. biology, chemistry. However, as noted on L34, we believe that “a critical component of automated science is automated reproducibility: the process of automatically reimplementing an experiment based on a description of the experiment design”. We will update the text to more clearly define our scope. Thanks for the suggestion.
>
>
> **[8M92-2] Performance degradation with richer hints?**
>
>
> We thank the reviewer for highlighting this important and surprising phenomenon. We would like to gently clarify that our results (see Sec. 3.3 and Table 3) showed *it was the o3-mini model, not DeepSeek-R1, whose performance tended to degrade when combining richer hints*. In contrast, DeepSeek-R1's performance was surprisingly benefitted by the additional context from combined hints (L186).
>
> As we detail in our response to Reviewer K2Q3 (see [K2Q3-4]), this behavior does not seem to be caused by exceeding the models' context limits, as all hints were designed to fit within the available context window. The more critical insight relates to how effectively each model can utilize the information it is given. For o3-mini, adding more sources of information appeared to complicate its task, suggesting a potential difficulty in reasoning over longer or more complex prompts. DeepSeek-R1, on the other hand, seemed to leverage the additional details from combined hints to better ground its implementation, leading to improved outcomes.
>
> This contrast in behavior highlights a key difference in the reasoning capabilities of these two frontier models. It is a crucial finding for future agentic scaffold design, as it suggests that the optimal strategy for providing information may need to be tailored to the specific language model being used. We will update the final text to reflect this.
>
>
> **[8M92-3] The metrics for code comparison are underdeveloped.**
>
>
> The primary goal of each task is to reproduce the functional outcome, i.e., the wall-clock speedup achieved by the human record which we measure directly with the Fraction of Speedup Recovered (FSR) metric. The exact syntax of the code is secondary. An agent could find a different, but equally effective, implementation of the same algorithmic idea, which should be considered a success.
>
>
> We included code similarity metrics (L2 code embedding distance in Fig. 5 and an R1-based judge score in Appendix C) as a secondary analysis to explore whether closeness to the human solution correlates with performance. We found a modest positive correlation, especially with richer hints, suggesting that these metrics can serve as a meaningful proxy for successful reproduction. However, we agree that developing robust methods for measuring semantic code similarity is a major research challenge in its own right, and one that is outside the primary scope of this paper, which is to introduce and validate the speedrunning benchmark itself.

---

> > ### Comment · Area_Chair_RXX3 · 2025-08-06
> >
> > Reviewer 8M92, can you acknowledge that you have red the rebuttal, and whether there is there anything that you would like to add to this discussion?

---

> > ### Comment · Reviewer_8M92 · 2025-08-06
> >
> > Thank you for solving my concerns. I raised my score

---

### Official Review · Reviewer_K2Q3 · 2025-07-03

**Rating:** 6
**Confidence:** 4

**Summary:**

The authors introduce "The Automated LLM Speedrunning Benchmark", a suite of 19 sequential tasks derived from the community‑driven NanoGPT Speedrun competition.   Each task asks an LLM‑based “research agent” to take the training script for record 𝐑ᵢ and reproduce the wall‑clock speed‑up achieved in the next record 𝐑ᵢ₊₁, optionally aided by three levels of “hints” that range from terse pseudocode to a mini‑paper.   The primary contribution is the benchmark itself, which offers a realistic, challenging, and non-saturated testbed for AI research agents. The authors evaluate agents based on state-of-the-art LLMs (DeepSeek-R1 and o3-mini) and find that even with detailed hints, these agents struggle to replicate the performance gains achieved by human experts, highlighting that automated reproducibility is a significant challenge for current AI systems. The paper is filled with interesting insights about the language models ability to accomplish tasks and it provides an incredible testbed to ground future experiments of LLM capabilities. A particularly impressive part of this paper is the detailed evaluation and search strategies employed by the authors to find a soft upper bound of what is possible with hints to accomplish a non trivial task that is highly relevant in the real world.

**Additional Feedback:**

NA

**Dataset Code Accessibility:**

Yes

**Dataset Code Comments:**

The authors shared all the records including the tweets of the findings and changes.

**Ethical Considerations:**

No, there are no or only very minor ethics concerns

**Final Justification:**

I am updating my score because I think this is a very valuable contribution to the field! With so many benchmarks that provide such little insights, this tasks provide so many new avenues of research. How good is an LLM in coming up with new optimizations? How good is it at following instructions? The instructions are for a novel task. Highly non-trivial. As someone who actively tries to integrate these novel changes into our codebases, I really hope the authors continue maintaining this benchmark for the community.

**Limitations Weaknesses:**

Even though the hints were manually verified they may not be perfect for the model’s comprehension abilities. Maybe github issues, discord chats, twitter posts that led to such a finding (not the finding itself, which the authors have already shared in the released artifact) or experiment would present the agent with a much more realistic scenario to come up with an effective solution. This is likely challenging to accomplish but raises the question that would some other form of instruction enable these agents to accomplish the tasks better?

The second weakness appears to be the hyper-parameter tuning component. The model might be able to come up with the solution but may not select the right hyper parameter.  Reporting both raw wall‑time and pass/fail correctness (e.g., reaches the target loss at all) would disentangle these effects. Also it would be great to see how quickly an average human can follow these hints and accomplish the task.

Lines 184 - 188 require a lot more explanation. If this is true, and context length limitation is significant then providing hints appropriate to context length is important.

**Strengths Contributions:**

There are many specific aspects about the paper that highlights the novelty of this work.

There has been a lot of hype around AI scientist type work and how LLMs are going to change fundamental research with lofty goals. However, lines 47 - 51 show that a task that should seemingly have been very easily solvable with the claims around AGI, is very far from being perfectly solved. This, accompanied with the fact that doing well in the speedrunning benchmark requires a sequential and cumulative decision path, something that two of the best reasoning models accomplished less than stellar performance, compared to humans (Albeit very smart humans who played a role in accomplishing the same goals).

The experimental design with progressive levels of hints along with different search methods mean the LLMs were tested in many possible configurations. This provides a great way to evaluate a soft upper bound on what is possible. H100 resources are highly limited and despite that the authors put a very sincere effort in presenting a very holistic evaluation framework.

The paper is very well-written. Easy to follow and fairly comprehensive. It also has many interesting nuggets of information like lines 169 - 174 about differences between R1’s approach and o3’s approach.

---

> ### Author Rebuttal · Authors · 2025-07-29
>
> We thank the reviewer for their positive assessment, highlighting the paper's novelty, the relevance of the task, the thoroughness of the evaluation, and the quality of the writing.
>
>
>
> **[K2Q3-1] Would other forms of instruction enable these agents to accomplish the tasks better?**
>
> This is an excellent point. A more "naturalistic" setting with noisier, unstructured data like GitHub issues or chat logs would indeed be a fascinating extension. We agree this could be a valuable direction for future work.
>
>
> However, we believe the current setup provides a crucial baseline. Our goal was to test the models' core reasoning and implementation capabilities under near-ideal conditions. The different forms of hints (i.e., instructions) we provide (pseudocode, text descriptions, and mini-papers) are manually verified distillations of the core code changes. The fact that models struggle even with these clean, targeted instructions suggests that providing more implicit or unstructured information from chat logs might not improve, and could even degrade, performance. Our experiment with adding the official FlexAttention documentation, which is a form of realistic external knowledge, supports this, as it surprisingly made performance worse for both models.
>
>
>
> **[K2Q3-2] Model may not select the right hyper parameters.**
>
>
> This is a key aspect of the challenge. The task is to reproduce a specific record, not to discover a new optimization from scratch. As such, our hint formats are designed to contain the necessary information to replicate the record, which includes relevant hyperparameter changes. Therefore, if a model fails to extract and apply the correct hyperparameters from the provided hints, this represents a failure in its ability to comprehend and follow instructions, the core capability our benchmark is designed to measure.
>
> To address your suggestion on reporting both raw wall-time and pass/fail correctness, our framework is designed to provide this separation. Our primary metric, the Fraction of Speedup Recovered (FSR), is directly calculated from raw wall-clock times, offering a standardized measure of performance improvement. For pass/fail correctness, our search tree analysis (Figure 6) provides a detailed breakdown, categorizing each attempt as a success ('Improved') or specifying the type of failure—be it a code crash ('Buggy') or a failure to improve performance or reach the validation loss target ('No Improvement'). These two components work together to provide a comprehensive view that disentangles the nature of the successes and failures.
>
>
>
>
> **[K2Q3-3] How quickly could an average human follow these hints to accomplish the task.**
>
> We agree this is a very interesting question. Establishing a baseline with human participants (who are not the original expert authors of the records) following the same hints would provide a valuable point of comparison for agent performance. This would be a significant user study, that would be non-trivial to conduct.  We believe it could be an excellent direction for future work to further contextualize the performance of AI agents.
>
>
>
> **[K2Q3-4] Providing hints appropriate to context length is important.**
>
> We appreciate the reviewer pointing this out for clarification. The issue observed on L187 is not about the hints exceeding the models' context windows. As we note in our limitations section on L272, all hint combinations were designed to be succinct and fit comfortably within the context length of both o3-mini and DeepSeek-R1.
>
> The critical finding here is about how effectively each model utilizes the provided context. The performance degradation of o3-mini when given more information (e.g., pseudocode + text + mini-paper vs. just pseudocode illustrated in Table 3) suggests a potential difficulty in reasoning over longer, more complex prompts, even when they fit within the context window. Conversely, R1's performance improvement with more context suggests it can better leverage richer prompts. This highlights a qualitative difference in the reasoning capabilities of the two models, which is a key insight from our study. We will update the final text to clarify this observation and the fact that hints fit within the context length.

---

> > ### Comment · Reviewer_K2Q3 · 2025-08-05
> > **Thank you for addressing the comments.**
> >
> > Thank you for addressing the comments. This is a worthy contribution to the field and will help us understand a lot about the capabilities of the models.

---

### Decision · Program_Chairs · 2025-09-18

**Decision:**

Accept (poster)

**Comment:**

This paper introduces “The Automated LLM Speedrunning Benchmark, a competition based on minimising the wall time of training an open-source reimplementation of GPT-2 to reach a target cross-entropy loss on the validation set of FineWeb, using a single 8xH100 node. Based on the reviews, it seems that the task is to train a sequential LLM agent that aims to solve the aforementioned task.

Most important strong points:
* Shows weaknesses of the LLM agents when it comes to aforementioned speedrunning tasks [R1, R3, R4]
* The introduced task is interesting and important: Utilisation of a community-driven benchmark to measure the ability to translate instructions into code [all]
* Extensive experimental evaluations using various configurations [R1]

Most important doubts:
* The result that even the strongest LLM agents have problems with long horizon tasks (~ mentioned by R1, context length limitation), might not be that surprising [R1, R3]
*The question generalisation beyond tasks of Speedrunning NanoGPT [R2, R3]

Other doubts that are mentioned only once or are not mission-critical, but would be good to reflect on:
* Doubts about the hint system [R1]
* Hyperparameter configuration neglected [R1]

Reason for acceptance: A very solid piece of work  (broad experimentation) that addresses an important topic. Reviewers were already positive before the discussion phase, and the discussion phase was relatively light-weight because of this (minor changes in scores).